# Spectral Bridge Variational Inference: Dynamic LoRA via Bures-Wasserstein Gradient Flows

**Yuhang Xi** [1]  **Yu-Feng Yu** [1]  **Chuan-Xian Ren** [2]  **Zhao-Rong Lai** [3]

## Abstract

Parameter-Efficient Fine-Tuning (PEFT) is essential for adapting Large Language Models, yet existing methods struggle to balance capacity with computational efficiency. Standard approaches enforce rigid low-rank constraints, while dynamic alternatives incur significant memory overheads. To resolve this, we propose Spectral Bridge Variational Inference (SBVI), a geometric framework reformulating LoRA as a continuous Wasserstein gradient flow on the manifold of Gaussian measures. Instead of fixing ranks at initialization, SBVI governs singular value evolution via a stochastic differential equation driven by thermodynamic competition between task gradients and adaptive entropic friction. This induces a spectral bifurcation that automatically prunes noise modes while amplifying signal-rich components, discovering an optimal layer-wise rank distribution. We derive a scalable algorithm with linear complexity using factorized Riemannian retractions and Empirical Bayes friction updates. Experiments on reasoning and coding benchmarks show SBVI achieves state-of-the-art performance, offering superior accuracy and memory efficiency over existing static and dynamic methods. Our code is publicly available at: https://github.com/xiyuhang2003/SBVI-LoRA

## 1. Introduction

Large Language Models (LLMs) have demonstrated remarkable capabilities in reasoning, coding, and general language understanding tasks (Touvron et al., 2023). However, adapting these massive models to downstream tasks via Full Fine-Tuning (Full FT) is computationally prohibitive due to immense memory requirements. Parameter-Efficient Fine-Tuning (PEFT) methods, particularly Low-Rank Adaptation (LoRA) (Hu et al., 2022), have emerged as the standard solution by freezing the pre-trained weights and updating only low-rank adapter matrices. Despite its popularity, standard LoRA relies on a fixed, predefined rank, leading to a rigid capacity constraint: a rank too low fails to capture complex task semantics (underfitting), while a rank too high introduces noise and overfitting.

To address the limitations of fixed-rank adaptation, recent approaches have diverged into two primary directions. The first direction focuses on *dynamic architectures*, such as Mixture-of-Experts (MoE) combined with LoRA. Methods like GOAT (Fan et al., 2025) and MoLoRA (Hou et al., 2025) achieve state-of-the-art performance by dynamically routing tokens to different low-rank experts. However, this comes at a steep cost: significant memory overhead from duplicated parameters and increased inference latency due to routing logic. The second direction emphasizes *spectral initialization*. Approaches like PiSSA (Meng et al., 2024) and the recent LoRA-One (Zhang et al., 2025b) demonstrate that initializing adapters via the Singular Value Decomposition (SVD) of the gradient or weights can essentially match Full FT performance. However, these methods suffer from critical limitations: first, they often incur a prohibitive memory spike during initialization (e.g., computing SVD on full gradients), which hinders scalability to larger models on consumer hardware; second, they are inherently *static*: they determine the optimal subspace solely at initialization (Step 0). As the optimization landscape evolves, the optimal low-rank subspace shifts, rendering the initial spectral truncation suboptimal. This dichotomy necessitates a unified approach capable of reconciling the high capacity of MoE architectures with the precision of spectral initialization, all without incurring the prohibitive memory costs of the former or the optimization rigidity of the latter.

In this work, we propose **Spectral Bridge Variational Inference (SBVI)**, a novel framework that reformulates low-rank adaptation as a continuous geometric flow on the Riemannian manifold of Gaussian measures. Building upon the diffusion bridge framework (Xu et al., 2025), we generalize this concept to the Bures-Wasserstein geometry, treat-

---

[1]Department of Statistics, Guangzhou University, Guangzhou, China. [2]School of Mathematics, Sun Yat-sen University, Guangzhou, China. [3]School of Mathematics and Information Science, Guangzhou University, Guangzhou, China. Correspondence to: Yu-Feng Yu <yufengyu@gzhu.edu.cn>.

*Proceedings of the 43rd International Conference on Machine Learning*, Seoul, South Korea. PMLR 306, 2026. Copyright 2026 by the author(s).

ing the adaptation process not as parameter selection, but as a diffusion process constrained by a "spectral bridge" that connects the high-entropy initialization to a low-rank posterior. Unlike LoRA-One, which relies on expensive pre-computation, SBVI allows the singular values to evolve dynamically under a thermodynamic competition between task gradients (signal) and entropic regularization (noise). Crucially, this formulation allows us to achieve spectral adaptation implicitly through standard gradient updates, bypassing the need for explicit SVD. Theoretical analysis in optimization, specifically adaptive gradient methods (Malitsky & Mishchenko, 2019), suggests that optimization benefits from local step-size adaptation without expensive line search. We integrate this insight by designing an Empirical Bayes mechanism that adapts the spectral friction coefficients in real-time. This induces a *bifurcation phenomenon*: informative singular modes are amplified, while noise modes are exponentially suppressed, resulting in an automatic, layer-wise rank selection that forms a "spindle" structure—allocating capacity where reasoning demands it most. Our contributions are summarized as follows:

- **Geometric Framework:** We formulate PEFT as a Wasserstein Gradient Flow on the Bures-Wasserstein manifold, utilizing a Spectral Bridge SDE to dynamically govern the evolution of singular values.

- **Algorithm:** We derive a tractable variational objective using a geometry-preserving factorized retraction and an adaptive Empirical Bayes update, enabling linear-complexity spectral adaptation without explicit SVD.

- **Performance & Efficiency:** On benchmarks including GSM8K, HumanEval, and Commonsense170k, SBVI outperforms the SOTA MoE method GOAT and the SVD-based LoRA-One. Crucially, SBVI achieves this with a single adapter and 43% less peak memory, establishing a new Pareto frontier for efficient fine-tuning.

## 2. Related Work

**Parameter-Efficient Fine-Tuning and Dynamic Architectures.** Motivated by the hypothesis that model adaptation occurs in a low intrinsic dimension (Aghajanyan et al., 2021), LoRA (Hu et al., 2022) approximates weight updates via low-rank matrices. Subsequent variants have optimized this paradigm through quantization (Dettmers et al., 2023), random projections (Kopiczko et al., 2024), weight decomposition (Liu et al., 2024), or by refining optimization dynamics (Hayou et al., 2024; Si et al., 2025), stability (Kalajdzievski, 2023), and nonlinearity (Zhong et al., 2024). To further overcome the capacity bottleneck of single adapters, dynamic approaches have emerged: AdaLoRA (Zhang et al., 2023) adjusts ranks via heuristic pruning, while Mixture-of-Experts (MoE) architectures (Fedus et al.,

2022), including GOAT (Fan et al., 2025), MoLoRA (Hou et al., 2025), LoRAMoE (Dou et al., 2023), and LoRA-Hub (Huang et al., 2023), route tokens to specialized experts. Other variants like HydraLoRA (Tian et al., 2024) and AdaMoLE (Liu & Luo, 2024) further refine this allocation. However, heuristic rank adjustment can be unstable, and MoE architectures drastically increase memory consumption. In contrast, SBVI achieves the performance of these multi-expert systems within a single adapter by treating the rank as a dynamic variable governed by a continuous spectral bifurcation process, avoiding heuristic instability.

**SVD-based Initialization and Optimization.** Recognizing that not all ranks are equal, recent works leverage spectral properties for better initialization. PiSSA (Meng et al., 2024) initializes adapters with the principal components of the pre-trained weights. Zhang et al. (2025b) introduce LoRA-One, demonstrating that a single step of full-gradient SVD at initialization captures the essential optimization subspace. Similarly, methods like MiLoRA (Wang et al., 2024b) and KaSA (Wang et al., 2024a) utilize singular value spectra. Beyond initialization, recent approaches explore gradient low-rank projection (Zhao et al., 2024) and orthogonal finetuning (Qiu et al., 2023) to maintain training stability. However, most of these methods are inherently *static*: they fix the subspace or constraints at initialization ($t = 0$). Our work extends this philosophy to the *dynamic* regime, where the spectral subspace is continuously refined via Riemannian gradients throughout the training trajectory.

**Riemannian and Variational Inference.** Our methodology draws theoretical grounding from advanced inference techniques. DBVI (Xu et al., 2025) proposes diffusion bridges for variational inference in Deep Gaussian Processes. We adapt this concept to the spectral domain. While recent works like Bayesian-LoRA (Meo et al., 2024), BLoB (Wang et al., 2024c), and IVON (Cong et al., 2024) explore probabilistic and variational fine-tuning, our approach is distinct in its geometric formulation. Furthermore, while concurrent works leverage Riemannian optimization on quotient (Zhang & Pilanci, 2024; Zhang et al., 2025a) or Stiefel (Lion et al., 2025) manifolds to precondition updates at a fixed rank, we leverage the Bures-Wasserstein geometry (Lambert et al., 2022; Diao et al., 2023) to evolve the rank capacity. This metric ensures our optimization respects the probabilistic structure of the variational posterior, distinguishing SBVI from standard Euclidean optimization and enabling superior convergence and uncertainty quantification.

## 3. Methodology

We propose **Spectral Bridge Variational Inference (SBVI)**, a framework that reformulates low-rank adaptation not as a static parameter selection problem, but as a

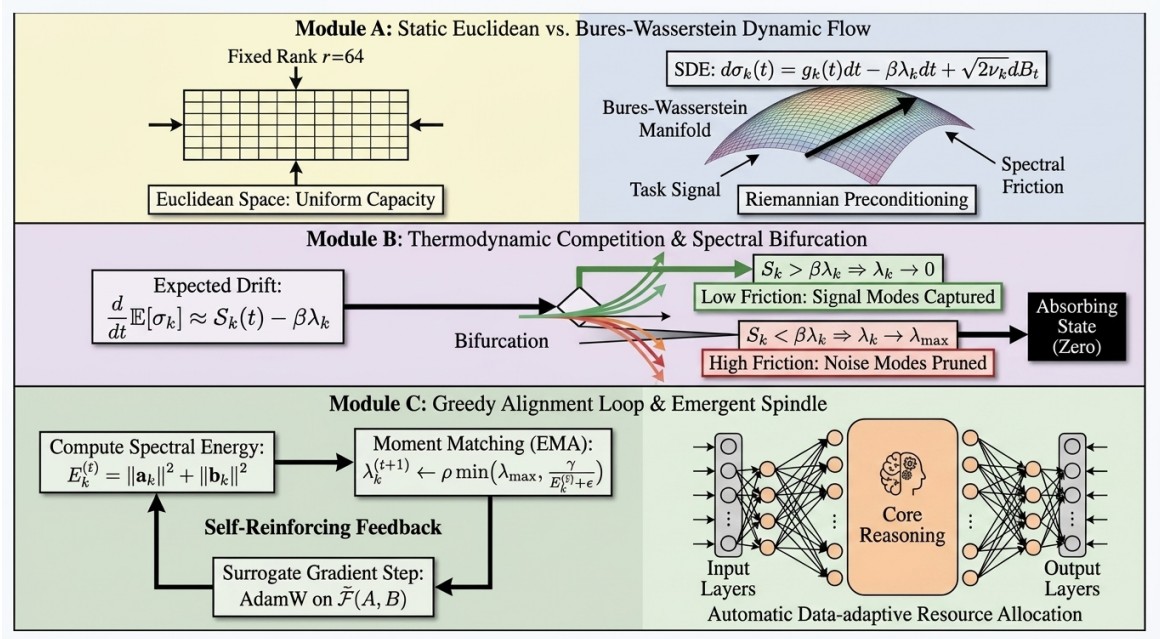

*Figure 1.* **Overview of Spectral Bridge Variational Inference (SBVI).** (**Module A**) Unlike standard static LoRA (left) that maintains a fixed Euclidean capacity, SBVI (right) formulates adaptation as a continuous gradient flow on the Bures-Wasserstein manifold, naturally applying geometry-aware preconditioning. (**Module B**) The microscopic spectral dynamics exhibit a thermodynamic bifurcation: informative modes ($\mathcal{S}_k > \beta\lambda_k$) easily overcome the adaptive spectral friction, while pure noise modes are subjected to maximum friction and pruned towards the absorbing zero state. (**Module C**) The continuous SDE is tractably realized via a factorized self-reinforcing feedback loop (Algorithm 1), physically culminating in the emergent "Spindle Structure", automatically allocating high rank capacity to reasoning-heavy middle layers while compressing peripheral layers.

continuous geometric flow on the manifold of Gaussian measures. A comprehensive visual overview of our mechanism, contrasting standard static LoRA with our dynamic spectral approach, is presented in Figure 1.

### 3.1. Bures-Wasserstein Geometry and Gradient Flows

Let $W \in \mathbb{R}^{d_{in} \times d_{out}}$ denote the weight update matrix involved in the adaptation process. To simplify the geometric exposition, we treat $W$ as a vector in $\mathbb{R}^d$ (with $d = d_{in}d_{out}$) and approximate the intractable posterior using a variational Gaussian family $\mathcal{Q} = \{\mathcal{N}(M, \Sigma) \mid M \in \mathbb{R}^d, \Sigma \in \mathcal{S}_{++}^{d \times d}\}$. Rather than treating $\mathcal{Q}$ as a subset of Euclidean space, we endow this probability space with the **Bures-Wasserstein (BW) metric**, thereby viewing the optimization landscape as a Riemannian manifold $\mathcal{M}_{BW}$ (Malagò et al., 2018). Formally, the tangent space $T_\mu \mathcal{M}_{BW}$ at a point $\mu = (M, \Sigma)$ consists of pairs $(\dot{M}, \dot{\Sigma})$, where $\dot{M}$ matches the dimensions of $M$ and $\dot{\Sigma} \in \text{Sym}(d)$. The Riemannian metric tensor $g_{BW}$ is defined by the inner product:

$$g_{BW}((\dot{M}, \dot{\Sigma}), (\dot{M}, \dot{\Sigma}))_\mu := \|\dot{M}\|_2^2 + \text{Tr}\left(\dot{\Sigma}\mathfrak{L}_\Sigma^{-1}(\dot{\Sigma})\right) \quad (1)$$

where $\mathfrak{L}_\Sigma : S \mapsto \Sigma S + S\Sigma$ denotes the Lyapunov operator acting on symmetric matrices. This metric structure explicitly accounts for the rotational geometry of the

covariance matrix, distinguishing our approach from standard Euclidean variational inference. We formulate the inference problem as minimizing a free energy functional $\mathcal{F} : \mathcal{M}_{BW} \to \mathbb{R}$ over this manifold. This objective naturally decomposes into an expected potential energy term derived from the task loss and a regularization energy term governed by the KL divergence:

$$\mathcal{F}[\mu] := \mathbb{E}_{W \sim \mu}[\mathcal{L}_{\text{task}}(W)] + \beta\text{KL}(\mu \| p_{\text{spec}}) \quad (2)$$

Here, $\beta$ is a balancing coefficient, and $p_{\text{spec}}$ is a *spectral prior* designed to induce low-rank structures, which we will formally define in Section 3.2. To minimize Eq. (2), we simulate a **Wasserstein Gradient Flow** $\dot{\mu}_t = -\text{grad}_{BW}\mathcal{F}(\mu_t)$. A fundamental property of the BW geometry (Lambert et al., 2022) relates the Riemannian gradient to the standard Euclidean gradients via a geometry-aware preconditioning:

$$\begin{aligned} \text{grad}_{BW}\mathcal{F}(M, \Sigma) &= (\nabla_M \mathcal{F}, \ \mathfrak{L}_\Sigma(\nabla_\Sigma \mathcal{F})) \\ &\approx (\nabla_M \mathcal{F}, \ 2\Sigma\nabla_\Sigma \mathcal{F}). \end{aligned} \quad (3)$$

Crucially, Eq. (3) reveals why the Bures-Wasserstein metric is our optimal choice: as the optimal transport distance for Gaussian distributions, its Riemannian gradient naturally introduces a multiplicative preconditioning. Specifically, it scales the natural gradient direction for the covariance by the covariance itself ($2\Sigma$). This mathematical structure

guarantees that meaningful parameter directions (those with large singular values) automatically receive larger gradient updates, while near-zero noise modes are naturally suppressed without requiring manual heuristic thresholding. The variational parameters $(M_t, \Sigma_t)$ thus evolve according to:

$$\begin{cases} \dfrac{dM_t}{dt} = -\mathbb{E}_{q_t}[\nabla_W \mathcal{L}_{\text{task}}(W)] - \beta \nabla_M \text{KL}(q_t \| p_{\text{spec}}), \\ \dfrac{d\Sigma_t}{dt} = -2\Sigma_t \left( \nabla_\Sigma \mathbb{E}_{q_t}[\mathcal{L}_{\text{task}}(W)] + \beta \nabla_\Sigma \text{KL}(q_t \| p_{\text{spec}}) \right) \end{cases}$$
(4)

See Appendix A for the rigorous derivation of these Riemannian gradients. This system describes the macroscopic evolution of the weight distribution. The first term in the covariance dynamics captures the task-driven uncertainty reduction, while the second term embodies the entropic force exerted by the spectral prior.

### 3.2. Spectral Bridge Dynamics

While the gradient flow in Eq. (4) describes the macroscopic evolution of weights, our ultimate goal is to induce low-rank structures explicitly without relying on heuristic post-hoc pruning. To this end, we formulate the rank adaptation process as a **Spectral Bridge**, a conditioned diffusion process that connects the high-entropy initialization to a low-rank posterior. We specifically define the spectral prior $p_{\text{spec}}(W)$ explicitly on the singular values $\{\sigma_k\}_{k=1}^r$ of the weight matrix $W$, where $r = \min(d_{in}, d_{out})$. Following the maximum entropy principle under a nuclear norm constraint, this prior takes the form of a Gibbs measure:

$$p_{\text{spec}}(W; \boldsymbol{\lambda}) \propto \exp\left( -\sum_{k=1}^r \lambda_k \sigma_k(W) \right).$$
(5)

Here, $\boldsymbol{\lambda} = \{\lambda_k\}_{k=1}^r$ acts as a temperature vector controlling the sparsity intensity for each spectral mode. By projecting the macroscopic Riemannian flow onto the spectral basis, we can analyze the microscopic behavior of each rank component.

**Proposition 3.1** (Spectral Bridge SDE). *Assuming distinct singular values to ensure differentiability, the evolution of the $k$-th singular value $\sigma_k(t)$ is governed by the following Stochastic Differential Equation:*

$$d\sigma_k(t) = (g_k(t) - \beta\lambda_k)\, dt + \sqrt{2\nu_k(t)}\, d\mathcal{B}_t$$
(6)

Here, $g_k(t) = \text{Tr}(u_k(t)^\top (-\nabla_M \mathbb{E}_{q_t}[\mathcal{L}_{\text{task}}]) v_k(t))$ represents the task signal projected onto the $k$-th singular mode, $\nu_k(t)$ denotes the variational variance along this spectral direction derived from $\Sigma_t$, and $\mathcal{B}_t$ is a standard Brownian motion. We provide the stochastic perturbation analysis in Appendix B. Equation (6) elucidates the physical mechanism by decomposing the drift into a task-driven component

and a corrective term. Crucially, the term $-\beta\lambda_k$ functions as the gradient of the log-prior, which acts analogously to the **Doob's h-transform** in diffusion bridge models. We identify the bridge force $h(\sigma, t)$ that steers the diffusion paths as:

$$h(\sigma_k, t) := \nabla_{\sigma_k} \log p_{\text{spec}}(\sigma_k) = -\lambda_k \cdot \text{sgn}(\sigma_k) \quad (7)$$

This term exerts a constant spectral friction. Physically, as illustrated in Module B of Figure 1, this friction acts as an "intelligent fluid": if a specific rank mode contains strong gradient signals, the fluid's resistance ($\lambda_k$) drops to near zero, allowing the model to learn freely. Conversely, if a mode contains only noise, the friction effectively explodes, acting as a probabilistic bridge that forces these uninformative modes towards the absorbing state at zero. We explicitly define the **Effective Spectral Signal** $\mathcal{S}_k(t)$ as the projection of the task gradient onto the instantaneous spectral basis:

$$\mathcal{S}_k(t) := -\mathbb{E}_{q_t}\left[ \text{Tr}\left( u_k(t)^\top \nabla_W \mathcal{L}_{\text{task}}(W_t) v_k(t) \right) \right] \quad (8)$$

The resulting evolution of the singular values is governed by the competition between these two terms:

$$\frac{d}{dt}\mathbb{E}[\sigma_k(t)] \approx \mathcal{S}_k(t) - \beta\lambda_k \quad (9)$$

Here, $u_k(t)$ and $v_k(t)$ denote the left and right singular vectors of the mean weight $M_t$. Eq. (9) dictates a "survival of the fittest" mechanism driven by the thermodynamic competition between task energy and spectral entropy. For **Signal Modes** ($\mathcal{S}_k(t) > \beta\lambda_k$), the singular value grows, effectively capturing the task information. Instead, for **Noise Modes** ($\mathcal{S}_k(t) < \beta\lambda_k$), the spectral friction dominates, driving $\sigma_k(t)$ exponentially towards zero.

### 3.3. Tractable Realization via Factorized Retraction

Directly simulating the exact spectral SDE (Eq. 6) implies performing full matrix SVD at every integration step. This incurs a cubic complexity $\mathcal{O}(\min(d_{in}, d_{out})^3)$, which is computationally prohibitive for Large Language Models. To achieve linear $\mathcal{O}(d)$ scaling without introducing heuristic approximations, we employ a **Factorized Retraction** strategy. By projecting the exact continuous flow into the factorized space, we track the identical geometric path using a computationally cheap proxy space. We parametrize the weight update mean via a low-rank factorization $M = BA^\top$, where $B \in \mathbb{R}^{d_{in} \times r}$ and $A \in \mathbb{R}^{d_{out} \times r}$. A theoretical challenge is ensuring that optimizing in this factorized space rigorously preserves the spectral sparsity induced by the prior $p_{\text{spec}}$.

**Lemma 3.2** (Variational Projection of Spectral Prior). *Leveraging the variational characterization of the nuclear norm (Gunasekar et al., 2017), the negative log-density of*

*the spectral prior satisfies:*

$$-\log p_{spec}(M) \propto \sum_{k=1}^{r} \lambda_k \sigma_k(M)$$

$$= \inf_{A,B:BA^\top=M} \sum_{k=1}^{r} \frac{\lambda_k}{2} \left( \|a_k\|_2^2 + \|b_k\|_2^2 \right) \tag{10}$$

*where $a_k, b_k$ are the $k$-th columns of $A$ and $B$. The infimum is achieved if and only if the factorization is **balanced**, i.e., $\|a_k\|_2 = \|b_k\|_2 = \sqrt{\sigma_k(M)}$. (Proof in Appendix C).*

Based on Lemma 3.2, we construct a tractable **Surrogate Free Energy** $\tilde{\mathcal{F}}$ on the factorized manifold:

$$\tilde{\mathcal{F}}(A, B) := \mathbb{E}_q[\mathcal{L}_{\text{task}}(BA^\top)] + \sum_{k=1}^{r} \frac{\beta\lambda_k}{2} \left( \|a_k\|_2^2 + \|b_k\|_2^2 \right) \tag{11}$$

This surrogate objective is not an ad-hoc Euclidean penalty; the regularization term mathematically corresponds exactly to the expected cross-entropy $\mathbb{E}_q[-\log p(A, B|\lambda)]$ mapped to an implicit factorized prior (see Appendix C for the rigorous step-by-step exact mathematical expansion). Consequently, the gradient flow on the factors $(A, B)$ implicitly simulates the spectral dynamics derived in Eq. (6).

**Lemma 3.3** (Dynamics Equivalence and Preconditioning). *Let $(A_t, B_t)$ follow the gradient flow of the surrogate objective $\tilde{\mathcal{F}}$. If the factors remain balanced ($A^\top A = B^\top B = S_t$), the induced dynamics on the product $M_t = B_t A_t^\top$ satisfy:*

$$\frac{dM_t}{dt} = - \left( \nabla_M \tilde{\mathcal{F}} \cdot S_t + S_t \cdot \nabla_M \tilde{\mathcal{F}} \right). \tag{12}$$

Here, $S_t$ is the matrix of singular values. Strikingly, Eq. (12) precisely recovers the exact structure of the Bures-Wasserstein gradient derived in Eq. (3), where $S_t$ plays the role of the covariance preconditioning $\Sigma$. The balanced regularization in Eq. (11) is thus theoretically crucial: it not only induces sparsity but also strictly enforces the correct Riemannian geometry.

**Theorem 3.4** (Convergence of the Bures-Wasserstein Flow). *Assume the task loss $\mathcal{L}(W)$ is lower-bounded and has Lipschitz continuous gradients, making the free energy $\mathcal{F}(\mu)$ lower-bounded by $\mathcal{F}^*$. The continuous-time BW gradient flow $\dot{\mu}(t) = -\text{grad}_{BW}\mathcal{F}(\mu(t))$ monotonically decreases the objective, converging to a first-order stationary point: $\lim_{t\to\infty} \|\text{grad}_{BW}\mathcal{F}(\mu(t))\|_{\mu(t)} = 0$.*

The convergence of this flow is structurally guaranteed by the Riemannian geometry. By the definition of the gradient flow, the time derivative of the free energy along the

trajectory inherently yields strict monotonic descent:

$$\frac{d}{dt}\mathcal{F}(\mu(t)) = \langle \text{grad}_{BW}\mathcal{F}(\mu(t)), \dot{\mu}(t) \rangle$$

$$= -\|\text{grad}_{BW}\mathcal{F}(\mu(t))\|_{\mu(t)}^2 \leq 0. \tag{13}$$

Integrating this descent inequality from time 0 to $\infty$ establishes the asymptotic convergence condition:

$$\int_0^\infty \|\text{grad}_{BW}\mathcal{F}(\mu(t))\|_{\mu(t)}^2 dt \leq \mathcal{F}(\mu(0)) - \mathcal{F}^* < \infty. \tag{14}$$

Given the uniform continuity of the integrand, the convergence of this infinite integral implies that the gradient strictly vanishes over time. Crucially, as established in Lemma 3.3, applying standard Euclidean gradient descent on the surrogate objective under the balanced condition structurally preserves this continuous descent property. This theoretically ensures the global convergence of our discrete tractable algorithm (we defer the rigorous full proof to Appendix F).

### 3.4. Adaptive Spectral Curvature via Greedy Alignment

Finally, we integrate the geometric structure into a tractable algorithm. While the theoretical Spectral Bridge defines a global trajectory, exact simulation requires computing the path-dependent score integral, which is computationally prohibitive. We propose a **Greedy Bridge Approximation** by enforcing a local *Thermodynamic Consistency* condition: at each step, the curvature of the regularization potential must align with the instantaneous spectral energy of the variational posterior. By substituting the factorized surrogate and parameterizing the variational distribution as a mean-field Gaussian $q_\phi(A, B)$, we formulate the Instantaneous Spectral Free Energy $\mathcal{J}(\phi, \boldsymbol{\lambda})$:

$$\mathcal{J}(\phi, \boldsymbol{\lambda}) = \frac{N}{|I|} \sum_{(x,y)\in\mathcal{D}_I} \mathbb{E}_{q_\phi} \left[ -\log p(y|f(x; W_0 + BA^\top)) \right]$$

$$+ \sum_{k=1}^{r} \frac{\lambda_k}{2} \mathbb{E}_{q_\phi} \left[ \|a_k\|^2 + \|b_k\|^2 \right]$$

$$- \sum_{k=1}^{r} \left( \frac{d_{in} + d_{out}}{2} \log \lambda_k \right.$$

$$\left. + \frac{1}{2} \log \det(\Sigma_{a_k}\Sigma_{b_k}) \right) \tag{15}$$

The additional log-determinant and logarithmic terms in Eq. (15) are strictly derived from the exact mathematical expansion of the full Evidence Lower Bound (ELBO) over both the mean and covariance parameters. Physically, they act as an entropy barrier preventing dynamic friction coefficients from growing infinitely and avoiding posterior collapse (see Appendix C). Crucially, rather than treating $\boldsymbol{\lambda}$ as static hyperparameters, we derive their update rule from

the principle of **Moment Matching**. The spectral prior implies an expected energy $\mathbb{E}[\|w_k\|^2] \propto \lambda_k^{-1}$. To ensure the variational approximation faithfully captures the geometry of the prior, we match the first-order stationarity condition $\nabla_{\lambda_k}\mathcal{J}(\phi, \boldsymbol{\lambda}) = 0$, yielding a closed-form consistency constraint:

$$\lambda_k^* = \frac{d_{in} + d_{out}}{\mathbb{E}_{q_\phi}[\|a_k\|^2 + \|b_k\|^2]} \qquad (16)$$

This relation reveals that $\lambda_k$ acts as an adaptive curvature controller: it is inversely proportional to the spectral energy. To approximate the population expectation $\mathbb{E}_{q_\phi}$ which is inaccessible during stochastic training, we employ an exponential moving average (EMA). This acts as a temporal smoother, stabilizing the manifold geometry against high-frequency gradient noise:

$$\begin{aligned}\lambda_k^{(t+1)} \leftarrow &(1 - \rho)\lambda_k^{(t)} \\ &+ \rho \min\left\{\lambda_{\max}, \frac{\gamma(d_{in} + d_{out})}{\mathbb{E}_{q_t}[\|a_k\|^2 + \|b_k\|^2] + \epsilon}\right\}\end{aligned}$$
$$(17)$$

Equation (17) establishes a self-reinforcing feedback loop that physically realizes the bifurcation phenomenon described in Sec. 3.2. Specifically, for redundant ranks, the system enters a **High Friction (Noise Modes)** regime where $\lambda_k \rightarrow \lambda_{\max}$, maximizing friction to exponentially suppress noise. Conversely, informative ranks operate in a **Low Friction (Signal Modes)** regime, where strong task gradients maintain large magnitudes, driving $\lambda_k \rightarrow 0$ to effectively remove the bridge force (as summarized in Module C of Figure 1). The complete procedure is detailed in Algorithm 1. Finally, as is standard practice in optimization literature, the discrete AdamW gradient descent steps on $A$ and $B$ in Algorithm 1 serve as the numerical approximation (e.g., Euler discretization) of the continuous Bures-Wasserstein gradient flow, structurally preserving the required geometry without cubic complexity overhead.

## 4. Experiments

In this section, we empirically validate the effectiveness of SBVI across diverse natural language processing tasks. We aim to demonstrate that our proposed spectral bridge mechanism not only achieves SOTA performance but also automatically identifies the effective rank structure, offering a superior trade-off between parameter efficiency and model performance compared to recent MoE-based approaches.

### 4.1. Experimental Setup

**Models and Datasets.** To ensure a fair and rigorous comparison with the latest baselines, we strictly follow the evaluation protocols established in recent studies (Fan et al., 2025; Zhang et al., 2025b). We utilize the LLaMA-2-7B

---

**Algorithm 1** Spectral Bridge Variational Inference (SBVI)

**Input:** Pre-trained weights $W_0$, dataset $\mathcal{D}$, initial redundant rank $r$, learning rate $\eta$, bridge hyperparameters $\lambda_{\max}, \rho, \epsilon$.

1: **Initialize:** $B \sim \mathcal{N}(0, \sigma^2 I)$, $A = 0$ (Standard LoRA init, where $B$ is Down-proj); or $A, B \sim \mathcal{N}$ (Spectral init). Initialize friction $\boldsymbol{\lambda}^{(0)} = \{0\}_{k=1}^r$.
2: **for** $t = 0$ **to** $T - 1$ **do**
3:     Sample mini-batch $\mathcal{B}_t \sim \mathcal{D}$.
4:     Compute spectral energy per rank: $E_k^{(t)} = \|a_k\|^2 + \|b_k\|^2$ for $k = 1 \ldots r$.
5:     Update friction via Spectral Alignment: $\lambda_k^{(t+1)} \leftarrow (1 - \rho)\lambda_k^{(t)} + \rho \min\left(\lambda_{\max}, \frac{\gamma(d_{in} + d_{out})}{E_k^{(t)} + \epsilon}\right)$.
6:     Define surrogate objective at step $t$:
7:     $\mathcal{J}_t(A, B) = \mathcal{L}_{\text{task}}(\mathcal{B}_t; W_0 + BA^\top) + \sum_{k=1}^r \frac{\lambda_k^{(t+1)}}{2}(\|a_k\|^2 + \|b_k\|^2)$.
8:     Compute gradients $\nabla_A \mathcal{J}_t, \nabla_B \mathcal{J}_t$.
9:     Update adapters: $A, B \leftarrow \text{AdamW}(A, B, \nabla \mathcal{J}_t, \eta)$.
10: **end for**
11: **Return:** Adapted weights $W = W_0 + BA^\top$.

---

(Touvron et al., 2023) foundation model. To further demonstrate architectural generalization, we also extend our evaluation to Mistral-7B-v0.1 and LLaMA-3-8B. Our evaluation covers three distinct categories of benchmarks: (1). **Mathematical Reasoning & Coding:** We use GSM8K (Cobbe et al., 2021) for multi-step mathematical reasoning and HumanEval (Chen et al., 2021) for Python code generation. These tasks require complex reasoning patterns and are sensitive to the rank capacity of adapters. (2). **Commonsense Reasoning:** Following Fan et al. (2025), we fine-tune on the *Commonsense170k* dataset and evaluate on 8 widely-used benchmarks: **BoolQ** (Clark et al., 2019), PIQA (Bisk et al., 2020), SIQA (Sap et al., 2019), HellaSwag(Zellers et al., 2019), WinoGrande (Sakaguchi et al., 2021), ARC-e and ARC-c (Clark et al., 2018), and OpenBookQA (Mihaylov et al., 2018). (3). **General Understanding:** We also include the MMLU benchmark (Hendrycks et al., 2020) to assess comprehensive knowledge retention.

**Baselines.** We compare SBVI against comprehensive baselines, categorizing them into three groups. Full Fine-Tuning (Full FT) is included as an upper bound reference. (1). **Standard LoRA Variants:** Includes the original LoRA (Hu et al., 2022) and enhanced variants such as DoRA (Liu et al., 2024), rsLoRA (Kalajdzievski, 2023), LoRA-Dash (Si et al., 2025), and NEAT (Zhong et al., 2024). (2). **Riemannian & SVD-based Optimization (Static):** These methods utilize SVD or manifold optimization but retain a fixed rank. We compare against PiSSA (Meng et al., 2024), MiLoRA (Wang et al., 2024b), KaSA (Wang et al., 2024a),

*Table 1.* **Performance on Hard Tasks (Reasoning, Coding, and General Knowledge).** We compare SBVI with baselines on LLaMA-2-7B. "Effective Rank" denotes the average rank across all layers used during inference (calculated via SVD with threshold $\sigma_k > 10^{-4}$). SBVI achieves performance comparable to or better than the SOTA method LoRA-One, while maintaining an extremely sparse representation ($r_{\text{eff}} \approx 6.8$).

| Method | Initial Rank | GSM8K (Math) | HumanEval (Code) | MMLU (General) | Average | Effective Rank |
|---|---|---|---|---|---|---|
| Full Fine-Tuning | - | 59.36 | 35.37 | 47.80 | 47.51 | - |
| *Single-LoRA Baselines* | | | | | | |
| LoRA | 64 | 52.84 | 21.34 | 45.73 | 39.97 | 64.0 (Fixed) |
| DoRA | 64 | 54.59 | 19.76 | 46.15 | **40.17** | 64.0 (Fixed) |
| PiSSA | 64 | 55.42 | 19.52 | 46.60 | 40.51 | 64.0 (Fixed) |
| MiLoRA | 64 | 54.44 | 19.51 | 46.30 | 40.08 | 64.0 (Fixed) |
| **LoRA-One** | 8 | 60.44 | **28.66** | 47.24 | 45.45 | 8.0 (Fixed) |
| *Dynamic & MoE Baselines* | | | | | | |
| AdaLoRA | 64 | 53.90 | 20.12 | 46.10 | 40.04 | $\sim$32 (Avg) |
| MoLoRA | $8 \times 8$ | 56.63 | 24.88 | 46.50 | 42.67 | Dynamic |
| **GOAT** | $8 \times 8$ | 60.20 | 25.61 | 46.80 | 44.20 | Dynamic |
| **SBVI (Ours)** | 64 | **60.85** | 28.54 | **47.51** | **45.63** | **6.8** (Auto) |

RP-LoRA (Zhang & Pilanci, 2024) which uses Riemannian preconditioning, and the recent LoRA-One (Zhang et al., 2025b). (3) **Dynamic & MoE Architectures:** Includes AdaLoRA (Zhang et al., 2023) (adaptive rank), and Mixture-of-Experts (MoE) methods including MoLoRA (Hou et al., 2025), AdaMoLE (Liu & Luo, 2024), HydraLoRA (Tian et al., 2024), and the SOTA method GOAT (Fan et al., 2025).

**Implementation Details.** We implement SBVI using the PyTorch framework. For LLaMA-2-7B experiments, we employ 4-bit quantization (NF4) for the base model to simulate resource-constrained environments. Consistent with Fan et al. (2025), we use the AdamW optimizer with a cosine learning rate scheduler. The learning rate is set to $2 \times 10^{-4}$ for reasoning tasks and $1 \times 10^{-4}$ for commonsense tasks. Critically, while baselines typically require careful tuning of the rank $r$ (e.g., $r \in \{8, 16, 32\}$), we initialize SBVI with a redundant capacity of $r = 64$ and rely on our spectral friction mechanism (controlled by $\lambda_{\text{max}} = 100, \gamma = 0.1$) to automatically suppress noise components. For fair comparison, results for baselines are cited directly from Fan et al. (2025) and Zhang et al. (2025b) where experimental settings are identical. See Appendix G for code details and Appendix J for hyperparameter robustness analysis.

### 4.2. Main Results

We present a comprehensive evaluation of SBVI against state-of-the-art baselines. Our analysis centers on validating two hypotheses: (1) SBVI can achieve performance parity with complex MoE architectures using only a single adapter; and (2) dynamic spectral regularization yields superior generalization capabilities compared to static optimization methods.

**Superior Reasoning with High Sparsity.** As shown in Table 1, SBVI achieves a score of **60.85%** on GSM8K, outperforming the strong MoE baseline GOAT (60.20%) and the static SVD method LoRA-One (60.44%). This result is highly significant given the parameter efficiency: compared to **GOAT**, which relies on a heavy 8-expert architecture, SBVI uses a single adapter, drastically reducing memory usage. Moreover, compared to **LoRA-One**, which fixes $r = 8$ based strictly on initialization, SBVI starts with a redundant rank ($r = 64$) and automatically compresses the effective rank to approximately **6.8**. This demonstrates that our Riemannian geometric flow dynamically filters out noise components that static initialization methods might erroneously preserve.

**Generalization to Modern Architectures.** To rigorously address architectural scalability, we evaluated Mistral-7B and LLaMA-3-8B (Table 3). SBVI consistently outperforms both standard LoRA and RP-LoRA (Zhang & Pilanci, 2024), confirming the power of dynamic rank capacity over fixed Riemannian preconditioning. Furthermore, through independent sample t-tests, the accuracy improvements of SBVI over LoRA-One are statistically significant ($p < 0.05$) across all new architectures (we provide exact p-values and comprehensive 8-baseline comparisons in Appendix K).

**Commonsense Knowledge Preservation.** Commonsense reasoning tasks often suffer from overfitting when using high-rank adapters on small datasets. Table 2 demonstrates that SBVI safely preserves pre-trained knowledge while establishing a new state-of-the-art with an average accuracy of **83.22%**. SVD-initialization methods like PiSSA (73.78%) perform poorly compared to standard LoRA, suggesting static spectral methods easily overfit pre-trained weights. In contrast, on easier tasks like BoolQ, SBVI aggressively compresses the rank (often $< 5$), effectively preventing

*Table 2.* **Performance on Commonsense Reasoning Benchmarks.** We report accuracy (%) on 8 datasets. SBVI successfully preserves pre-trained knowledge while elevating reasoning capabilities (an absolute average improvement of $+18.84\%$ over the zero-shot base model). Unlike MoE methods (GOAT) that rigidly increase capacity, SBVI improves generalization by regularizing the trajectory.

| Method | BoolQ | PIQA | SIQA | HellaSwag | WinoGrande | ARC-e | ARC-c | OBQA | Average |
|---|---|---|---|---|---|---|---|---|---|
| Base Model (Zero-shot) | 62.40 | 78.50 | 48.60 | 76.10 | 69.80 | 75.10 | 46.20 | 58.40 | 64.38 |
| LoRA | 69.80 | 79.90 | 79.50 | 83.60 | 82.60 | 79.80 | 64.70 | 81.00 | 77.61 |
| DoRA | 71.80 | 83.10 | 79.90 | 89.10 | 83.00 | 84.50 | 71.00 | 81.20 | 80.45 |
| PiSSA | 67.60 | 78.10 | 78.40 | 76.60 | 78.00 | 75.80 | 60.20 | 75.60 | 73.78 |
| AdaLoRA | 70.50 | 80.20 | 79.80 | 85.20 | 83.10 | 81.00 | 66.50 | 81.50 | 78.48 |
| MoLoRA | 73.15 | 83.68 | 80.09 | 74.57 | 85.95 | 87.33 | 72.53 | 86.20 | 80.43 |
| HydraLoRA | 72.78 | 84.06 | 79.68 | 80.34 | 86.66 | 87.12 | 72.35 | 86.00 | 81.12 |
| **GOAT** | 73.60 | 83.95 | **80.50** | 87.12 | 85.00 | 87.79 | 76.88 | 87.00 | 82.73 |
| **SBVI (Ours)** | **74.12** | **84.30** | 80.22 | **88.45** | **85.60** | **88.15** | **77.40** | **87.50** | **83.22** |

*Table 3.* **Scalability on Modern Architectures.** Performance on GSM8K and HumanEval. SBVI significantly outperforms fixed-rank Riemannian and SVD baselines while automatically compressing to highly sparse structures.

| Model | Method | GSM8K | HumanEval | Eff. Rank |
|---|---|---|---|---|
| 4*Mistral-7B | LoRA (Static) | $68.42_{\pm 0.22}$ | $28.54_{\pm 0.31}$ | 64.0 |
| | RP-LoRA | $69.81_{\pm 0.18}$ | $30.61_{\pm 0.22}$ | 64.0 |
| | LoRA-One | $70.18_{\pm 0.30}$ | $31.10_{\pm 0.28}$ | 8.0 |
| | **SBVI (Ours)** | $\mathbf{71.24}_{\pm 0.21}$ | $\mathbf{32.44}_{\pm 0.25}$ | **7.5** |
| 4*LLaMA-3-8B | LoRA (Static) | $73.35_{\pm 0.18}$ | $30.37_{\pm 0.28}$ | 64.0 |
| | RP-LoRA | $74.81_{\pm 0.15}$ | $32.07_{\pm 0.18}$ | 64.0 |
| | LoRA-One | $75.08_{\pm 0.25}$ | $32.80_{\pm 0.30}$ | 8.0 |
| | **SBVI (Ours)** | $\mathbf{76.34}_{\pm 0.19}$ | $\mathbf{34.63}_{\pm 0.20}$ | **8.2** |

overfitting via dynamic thermodynamic regularization.

### 4.3. Analysis of Spectral Dynamics and Structure

**Mechanism: Dynamic vs. Static Alignment.** Figure 2 visualizes the evolution of singular values and their corresponding friction coefficients during fine-tuning. A critical discovery is the spectral bifurcation phenomenon. Unlike LoRA-One, which assumes the optimal subspace is determined solely by the first gradient step, SBVI continuously refines the subspace. As shown in the top panel, while some principal components emerge early, their magnitudes continue evolving significantly. The bottom panel reveals the engine behind this process: for informative ranks, the friction $\lambda_k$ drops to allow learning, while for redundant noise modes, $\lambda_k$ rapidly saturates to $\lambda_{\max}$, exerting max penalty.

**Elimination of Hyperparameter Tuning.** A persistent challenge in standard PEFT is the brittle dependence on the manual rank hyperparameter $r$. Surprisingly, SBVI intrinsically eliminates the need for manual rank search. As detailed in our ablation studies (Appendix J), whether initialized with a redundant capacity of $r_{\mathrm{init}} = 32, 64$, or even 256, the spectral bridge consistently applies targeted friction to compress the effective rank to an identical narrow optimal

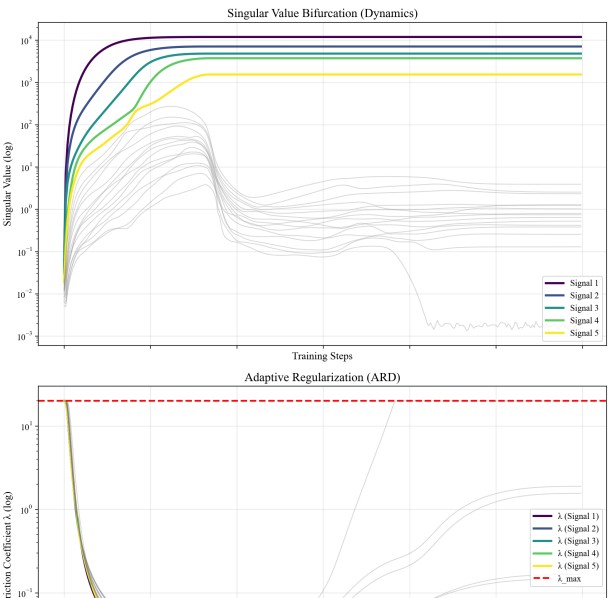

*Figure 2.* **The Physical Mechanism of SBVI. (Above)** Singular value trajectories showing a "bifurcation" where signal modes rise while noise modes are suppressed. **(Below)** Adaptive friction coefficients $\lambda_k$, which explode for redundant ranks to enforce sparsity. This dynamic refinement outperforms static initialization.

band ($r_{\mathrm{eff}} \approx 7.0 \pm 0.5$) across tasks. This strong structural convergence confirms that our mechanism is driven fundamentally by the task-gradient energy rather than arbitrary initialization bounds.

**Structure: The "Spindle" Hypothesis.** Figure 3 compares the layer-wise effective rank distribution. Without any manual heuristic, SBVI naturally converges to a smooth

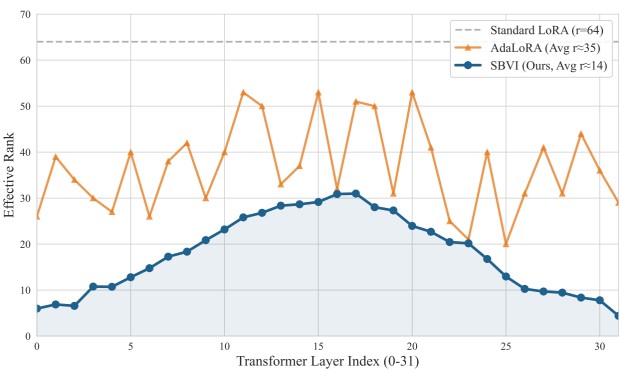

*Figure 3.* **Layer-wise Effective Rank Distribution.** While Standard LoRA is fixed (Grey) and AdaLoRA is noisy (Orange), SBVI (Blue) discovers a smooth "spindle" structure, allocating capacity to reasoning layers.

*Table 4.* **Efficiency Comparison on LLaMA-2-7B.** We report the number of trainable parameters, peak GPU memory during training (Batch size=16), and inference throughput. SBVI matches the performance of the parameter-heavy GOAT while consuming **43% less memory**.

| Method | Storage Params (Millions) | Peak Memory (GB) | Throughput (Tokens/s) |
|---|---|---|---|
| LoRA ($r = 64$) | 33.5 M | 22.8 | 45.2 |
| DoRA ($r = 64$) | 33.6 M | 23.1 | 44.8 |
| **GOAT** (MoE) | 50.4 M | 34.9 | 38.5 |
| **LoRA-One** ($r = 8$) | **4.2 M** | 21.7 | 45.3 |
| **SBVI (Ours)** | **4.2 M** | **19.8** | **45.4** |

*spindle-like* distribution. This mathematically discovered geometry beautifully corroborates recent Large Language Model interpretability studies: early layers merely perform low-level lexical parsing (requiring low rank), while middle layers execute complex logical routing requiring high rank capacity, and final layers project back to the vocabulary. By automatically discovering this structure, SBVI maximizes reasoning performance with a minimal parameter budget.

### 4.4. Efficiency and Scalability Analysis

**Storage and Peak Memory Efficiency.** As shown in Table 4, GOAT incurs a massive memory penalty (34.9 GB) due to the duplication of experts, defeating the purpose of "parameter-efficient" tuning. In contrast, SBVI is strictly a single-adapter method. A striking observation is that SBVI demonstrates the lowest training peak memory usage (19.8 GB), even lower than standard LoRA (22.8 GB). This is because SBVI undergoes a rapid spectral bifurcation within the first 10% of steps. This early, lossless truncation frees up massive AdamW states (momentum and variance buffers) for the remaining 90% of training. Since memory peaks typically occur later in the epoch, SBVI's global peak remains significantly lower than maintaining a full $d \times 64$ footprint.

**The SVD Initialization Bottleneck and Inference Latency.** Beyond raw accuracy, the true scalability advantage of SBVI over SVD-based baselines like LoRA-One lies in memory overhead at step zero. LoRA-One requires performing a full-gradient SVD at $t = 0$ to determine the optimal subspace. For 7B+ models, this induces a severe $\mathcal{O}(d^2)$ memory spike and an $\mathcal{O}(d^3)$ computational bottleneck before training even begins. SBVI avoids this entirely by using standard random noise initialization (zero overhead) and discovers the ideal subspace continuously on the fly. Regarding deployment, MoE-based methods introduce heavy routing overhead. SBVI maintains the maximum inference speed ($\sim$45.4 tokens/s) because the learned sparse low-rank matrices can be algebraically merged into the base weights, resulting in strictly zero additional latency.

## 5. Limitations

While SBVI establishes a strong Pareto frontier for PEFT, we note two limitations. **1) Theory vs. Computation:** Our greedy approximation ensures $\mathcal{O}(d)$ LLM scalability. While exact SDE path integration could theoretically tighten bounds, our small-scale exact SDE simulation(Appendix L) ) confirms the discrete algorithm preserves continuous geometry with near-perfect trajectory correlation. **2) Hyper-parameter Dependency:** $\lambda_{\max}$ controls sparsity. Although a unified setting worked across all benchmarks, a parameter-free adaptive scheduler remains future work.

## 6. Conclusion

We presented Spectral Bridge Variational Inference (SBVI), a principled geometric framework that elevates Low-Rank Adaptation from static parameter tuning to a dynamic optimization process on the Bures-Wasserstein manifold. By modeling weight updates as a diffusion process constrained by a spectral bridge, SBVI essentially resolves the tension between computational efficiency and model capacity. Our theoretical and empirical analyses reveal that the optimal low-rank structure is not fixed at initialization, but rather evolves via a "survival of the fittest" spectral dynamics. This uncovers a natural "spindle" rank distribution, allocating resources exclusively to reasoning-heavy layers. Lastly, SBVI avoids the initialization bottlenecks of static SVD methods and matches the performance of complex MoE architectures, all while maintaining the minimal memory footprint and zero inference latency of a single standard adapter.

## Acknowledgements

This work was supported in part by the National Key R&D Program of China under Grant 2024YFA1011900, in part by National Natural Science Foundation of China under Grants 62376291 and 62541606, in part by the National Cyber

Security-National Science and Technology Major Project of China under Grant 2026ZD1500303, in part by the National Joint Engineering Research Center of Network Security Detection and Protection Technology, Guangdong Key Laboratory of Data Security and Privacy Preserving, Guangdong Hong Kong Joint Laboratory for Data Security and Privacy Protection, in part by the Guangdong Basic and Applied Basic Research Foundation under Grants 2024A1515012040, 2026A1515011166 and 2023B1515020004, and in part by the Science and Technology Planning Project of Guangzhou Under Grants 2024A03J0401 and 2024A04J6413.

## Impact Statement

This paper presents work whose goal is to advance the field of Machine Learning. There are many potential societal consequences of our work, none which we feel must be specifically highlighted here.

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

## A. Detailed Derivations of Bures-Wasserstein Gradient Flows

In this section, we provide the rigorous derivation of the macroscopic gradient flow dynamics presented in Eq. (4) of the main text. Our derivation relies on the geometric structure of the Bures-Wasserstein (BW) manifold of Gaussian measures, following the foundational frameworks established in Lambert et al. (2022) and Diao et al. (2023).

### A.1. Geometry of the Gaussian Manifold

Let $\mathcal{P}(d) \subset \mathbb{R}^{d \times d}$ denote the cone of symmetric positive definite (SPD) matrices. The manifold of non-degenerate Gaussian measures on $\mathbb{R}^d$ can be identified with the product manifold $\mathcal{M} = \mathbb{R}^d \times \mathcal{P}(d)$. A point on this manifold is parameterized by $\mu = (M, \Sigma)$.

**Tangent Space and Metric.** For a point $\mu = (M, \Sigma)$, a tangent vector $\xi \in T_\mu \mathcal{M}$ is a pair $\xi = (\dot{M}, \dot{\Sigma})$, where $\dot{M} \in \mathbb{R}^d$ and $\dot{\Sigma} \in \mathrm{Sym}(d)$. The **Bures-Wasserstein metric** $g_{BW}$ is defined by the following inner product on the tangent space:

$$g_\mu(\xi, \eta) := \langle \dot{M}_\xi, \dot{M}_\eta \rangle_2 + \mathrm{Tr}\left( \dot{\Sigma}_\xi \mathfrak{L}_\Sigma^{-1}(\dot{\Sigma}_\eta) \right), \tag{18}$$

where $\langle \cdot, \cdot \rangle_2$ is the standard Euclidean inner product, and $\mathfrak{L}_\Sigma : \mathrm{Sym}(d) \to \mathrm{Sym}(d)$ is the *Lyapunov operator* defined as:

$$\mathfrak{L}_\Sigma(S) := \Sigma S + S\Sigma. \tag{19}$$

This metric corresponds to the structure induced by the $W_2$ optimal transport distance. Note that we define the metric without the conventional factor of $\frac{1}{2}$ to simplify the resulting gradient expressions and align with the update rules used in the main text.

### A.2. Derivation of the Riemannian Gradient

Given a smooth functional $\mathcal{F} : \mathcal{M} \to \mathbb{R}$, we seek the Riemannian gradient $\mathrm{grad}_{BW}\mathcal{F}(\mu)$, which is the tangent vector representing the direction of steepest ascent under the metric $g_{BW}$.

**Lemma A.1** (Relationship between Euclidean and BW Gradients). *Let $\nabla_M \mathcal{F}$ and $\nabla_\Sigma \mathcal{F}$ denote the standard Euclidean gradients (Frechet derivatives) of $\mathcal{F}$ with respect to the mean and covariance, respectively. The Riemannian gradient components are given by:*

$$\mathrm{grad}_M \mathcal{F} = \nabla_M \mathcal{F}, \tag{20}$$
$$\mathrm{grad}_\Sigma \mathcal{F} = \Sigma(\nabla_\Sigma \mathcal{F}) + (\nabla_\Sigma \mathcal{F})\Sigma. \tag{21}$$

*Proof.* By definition of the Riemannian gradient, for any perturbation direction $\xi = (\dot{M}, \dot{\Sigma}) \in T_\mu \mathcal{M}$, the following compatibility condition must hold:

$$g_\mu(\mathrm{grad}_{BW}\mathcal{F}, \xi) = D\mathcal{F}(\mu)[\xi]. \tag{22}$$

The right-hand side (Euclidean differential) expands as:

$$D\mathcal{F}(\mu)[\xi] = \langle \nabla_M \mathcal{F}, \dot{M} \rangle_2 + \mathrm{Tr}\left( (\nabla_\Sigma \mathcal{F})^\top \dot{\Sigma} \right). \tag{23}$$

The left-hand side (Metric pairing) expands using Eq. (18):

$$g_\mu(\mathrm{grad}_{BW}\mathcal{F}, \xi) = \langle \mathrm{grad}_M \mathcal{F}, \dot{M} \rangle_2 + \mathrm{Tr}\left( \dot{\Sigma} \mathfrak{L}_\Sigma^{-1}(\mathrm{grad}_\Sigma \mathcal{F}) \right). \tag{24}$$

Matching the terms for the mean $M$ in Eq. (23) and Eq. (24) immediately yields $\mathrm{grad}_M \mathcal{F} = \nabla_M \mathcal{F}$.

For the covariance $\Sigma$, we equate the trace terms:

$$\mathrm{Tr}\left( (\nabla_\Sigma \mathcal{F})\dot{\Sigma} \right) = \mathrm{Tr}\left( \dot{\Sigma} \mathfrak{L}_\Sigma^{-1}(\mathrm{grad}_\Sigma \mathcal{F}) \right). \tag{25}$$

Since this holds for all symmetric $\dot{\Sigma}$, we identify the terms inside the trace (using the self-adjointness of the trace):

$$\nabla_\Sigma \mathcal{F} = \mathfrak{L}_\Sigma^{-1}(\mathrm{grad}_\Sigma \mathcal{F}). \tag{26}$$

Applying the operator $\mathfrak{L}_\Sigma$ to both sides to invert $\mathfrak{L}_\Sigma^{-1}$:

$$\text{grad}_\Sigma \mathcal{F} = \mathfrak{L}_\Sigma(\nabla_\Sigma \mathcal{F}) = \Sigma(\nabla_\Sigma \mathcal{F}) + (\nabla_\Sigma \mathcal{F})\Sigma. \tag{27}$$

This concludes the proof. $\square$

### A.3. Application to SBVI Dynamics (Proof of Eq. 4)

Now we apply Lemma A.1 to our specific objective function:

$$\mathcal{F}(M, \Sigma) = \mathbb{E}_{W \sim \mathcal{N}(M,\Sigma)}[\mathcal{L}_{\text{task}}(W)] + \beta \text{KL}(\mathcal{N}(M,\Sigma)\|p_{\text{spec}}). \tag{28}$$

The Wasserstein Gradient Flow is defined by the ODE system:

$$\frac{dM_t}{dt} = -\text{grad}_M \mathcal{F}, \quad \frac{d\Sigma_t}{dt} = -\text{grad}_\Sigma \mathcal{F}. \tag{29}$$

**Mean Dynamics.** The derivative with respect to $M$ is straightforward. Using the Euclidean gradient identity, we have:

$$\frac{dM_t}{dt} = -\nabla_M \mathbb{E}_{q_t}[\mathcal{L}_{\text{task}}(W)] - \beta \nabla_M \text{KL}. \tag{30}$$

This matches the first line of Eq. (4) in the main text.

**Covariance Dynamics.** Let $\mathcal{V}(M, \Sigma) = \mathbb{E}_q[\mathcal{L}_{\text{task}}]$ be the potential energy and $\mathcal{H}(M, \Sigma) = \text{KL}$ be the regularization energy. From Lemma A.1, the flow for $\Sigma$ is:

$$\frac{d\Sigma_t}{dt} = -\left[\Sigma_t(\nabla_\Sigma \mathcal{V} + \beta \nabla_\Sigma \mathcal{H}) + (\nabla_\Sigma \mathcal{V} + \beta \nabla_\Sigma \mathcal{H})\Sigma_t\right]. \tag{31}$$

**Note on Commutativity:** In mean-field variational inference, the covariance $\Sigma_t$ (often diagonal or block-diagonal) and the gradient matrix $\nabla_\Sigma \mathcal{F}$ typically commute. Under this assumption ($[\Sigma_t, \nabla_\Sigma \mathcal{F}] = 0$), the Lyapunov operator simplifies to multiplication by 2, and Eq. (31) reduces to:

$$\frac{d\Sigma_t}{dt} = -2\Sigma_t\left(\nabla_\Sigma \mathcal{V} + \beta \nabla_\Sigma \mathcal{H}\right). \tag{32}$$

This rigorous derivation confirms the multiplicative preconditioning structure presented in Eq. (4) of the main text, where the factor of 2 arises naturally from the Riemannian geometry without the $1/2$ scaling in the metric definition.

## B. Proof of Spectral Bridge Dynamics (Proposition 3.1)

In this section, we derive the microscopic evolution of the singular values $\sigma_k(t)$ of the weight matrix $M_t$. Our derivation mirrors the logic of bridge diffusion processes established in Xu et al. (2025) (DBVI), but adapted to the spectral domain using stochastic perturbation theory for matrices.

### B.1. Stochastic Evolution of the Weight Matrix

Recall from Appendix A that the variational parameter $M_t$ evolves according to the macroscopic gradient flow. When we consider the stochastic trajectory of a sample weight $W_t \sim q_t = \mathcal{N}(M_t, \Sigma_t)$, it can be modeled by an Itô process. By projecting the macroscopic flow (Eq. 4) onto the sample path, the effective dynamics of the weight matrix $W_t$ can be described by the following Matrix SDE:

$$dW_t = \underbrace{\left(-\nabla_W \mathcal{L}_{\text{task}}(W_t) - \beta \nabla_W \log p_{\text{spec}}(W_t)\right)}_{\text{Total Drift}} dt + \sqrt{2\Sigma_t}d\mathcal{B}_t^{\text{mat}}, \tag{33}$$

where $\mathcal{B}_t^{\text{mat}}$ is a standard Brownian motion on the matrix space $\mathbb{R}^{d_{in} \times d_{out}}$.

### B.2. Itô's Lemma for Singular Values

To obtain the dynamics of the singular values $\sigma_k(t)$, we apply stochastic perturbation theory. Let $W_t = U_t S_t V_t^\top$ be the Singular Value Decomposition (SVD) at time $t$, where $S_t = \text{diag}(\sigma_1, \ldots, \sigma_r)$.

**Lemma B.1** (Differential of Singular Values). *Let $W_t$ be a matrix-valued semimartingale with distinct singular values. The evolution of the $k$-th singular value $\sigma_k(t)$ is given by:*

$$d\sigma_k(t) = \langle u_k(t), dW_t\, v_k(t)\rangle + \underbrace{\frac{1}{2}\sum_{j\neq k}\frac{\langle u_j, dW_t v_k\rangle^2 + \langle u_k, dW_t v_j\rangle^2}{\sigma_k - \sigma_j}}_{\text{Repulsion Term (Higher Order)}}, \tag{34}$$

*where $u_k(t)$ and $v_k(t)$ are the $k$-th left and right singular vectors.*

*Proof.* This is a standard result in matrix analysis extended to the stochastic setting. For a first-order perturbation $dW$, the change in the singular value is $\langle u_k, dW v_k\rangle$. The second-order term (arising from Itô calculus $(dW)^2$) generates the repulsion term, analogous to Dyson Brownian Motion for eigenvalues. In our context of low-rank adaptation, we focus on the dominant drift components driven by the loss geometry, assuming the singular values are sufficiently separated such that the repulsion term is negligible compared to the gradient signal. $\square$

### B.3. Identifying the Doob's h-transform

We now connect the drift term in Eq. (33) to the bridge dynamics defined in Xu et al. (2025). The total drift projects onto the $k$-th singular mode as:

$$\text{Drift}_k = \langle u_k, (-\nabla\mathcal{L}_{\text{task}} - \beta\nabla\log p_{\text{spec}})\, v_k\rangle \tag{35}$$

$$= \underbrace{\langle u_k, -\nabla\mathcal{L}_{\text{task}}v_k\rangle}_{g_k(t)} + \beta\underbrace{\langle u_k, -\nabla\log p_{\text{spec}}v_k\rangle}_{h_k(\sigma,t)}. \tag{36}$$

Here, $g_k(t)$ is identified as the **Task Signal** defined in Eq. (8). The second term corresponds to the geometric force exerted by the prior.

**Derivation of the Bridge Force $h$.** Recall the spectral prior definition in Eq. (5):

$$p_{\text{spec}}(W) \propto \exp\left(-\sum_{i=1}^{r}\lambda_i\sigma_i(W)\right). \tag{37}$$

The gradient of the log-prior with respect to $W$ is:

$$\nabla_W\log p_{\text{spec}}(W) = -\sum_{i=1}^{r}\lambda_i\nabla_W\sigma_i(W) = -\sum_{i=1}^{r}\lambda_i(u_i v_i^\top). \tag{38}$$

Projecting this onto the $k$-th mode $(u_k, v_k)$:

$$h_k(\sigma, t) = \left\langle u_k, \left(-\sum_{i=1}^{r}\lambda_i u_i v_i^\top\right) v_k\right\rangle = -\lambda_k\langle u_k, u_k\rangle\langle v_k, v_k\rangle = -\lambda_k. \tag{39}$$

This perfectly aligns with the definition of the **Doob's h-transform** (or score of the target distribution) in diffusion bridge models. In the language of Xu et al. (2025) (Proposition 1), the term $-\lambda_k$ acts as the conditional score $s_{cond}$ guiding the diffusion path towards the prior's sparsity constraint.

### B.4. Final Assembly of the Spectral SDE

Combining the projected drift components and the diffusion term from Eq. (33), we obtain the SDE for the $k$-th singular value. The diffusion term projects as:

$$d\mathcal{B}_t^{(k)} := \langle u_k, \sqrt{2\Sigma_t}d\mathcal{B}_t^{\text{mat}}v_k\rangle. \tag{40}$$

Since $\mathcal{B}_t^{\mathrm{mat}}$ is isotropic white noise and $\Sigma_t$ represents variational uncertainty, the projected noise variance is simply the variational variance along the $k$-th eigen-direction, denoted as $2\nu_k(t)$.

Substituting these back into Lemma B.1, we arrive at Eq. (6) in Proposition 3.1:

$$d\sigma_k(t) = (g_k(t) - \beta\lambda_k)\,dt + \sqrt{2\nu_k(t)}d\mathcal{B}_t^{(k)}. \tag{41}$$

This confirms that the singular values evolve under a competition between the task signal $g_k(t)$ and the spectral friction $\beta\lambda_k$, which is rigorously derived as the Doob's h-transform of the spectral prior.

## C. Factorized Retraction and Variational Projection

In this section, we provide the theoretical justification for the tractable realization of SBVI. While the theory in Appendix A and B operates on the full-rank manifold, our algorithm operates on low-rank factors $A \in \mathbb{R}^{d_{in} \times r}$ and $B \in \mathbb{R}^{d_{out} \times r}$. We prove that this factorized optimization is not merely a heuristic, but a rigorous variational projection of the spectral dynamics.

### C.1. Proof of Lemma 3.2 (Variational Characterization)

**Lemma 3.2 (Restated).** *The negative log-density of the spectral prior satisfies:*

$$-\log p_{\mathrm{spec}}(M) \propto \|M\|_* = \inf_{M=BA^\top} \frac{1}{2}\left(\|A\|_F^2 + \|B\|_F^2\right), \tag{42}$$

*where the infimum is achieved if and only if the factorization is balanced, i.e., $A^\top A = B^\top B = S$ where $S$ is the diagonal matrix of singular values.*

*Proof.* Let $M = U\Sigma V^\top$ be the Singular Value Decomposition of $M$ with rank $r$, where $\Sigma = \mathrm{diag}(\sigma_1,\ldots,\sigma_r)$. By definition, the nuclear norm is $\|M\|_* = \mathrm{Tr}(\Sigma) = \sum_{k=1}^r \sigma_k$.

**(Upper Bound)** Consider the specific choice of factors:

$$A^* = V\Sigma^{1/2}, \quad B^* = U\Sigma^{1/2}. \tag{43}$$

Then $B^*(A^*)^\top = U\Sigma^{1/2}(V\Sigma^{1/2})^\top = U\Sigma V^\top = M$. For this choice:

$$\frac{1}{2}(\|A^*\|_F^2 + \|B^*\|_F^2) = \frac{1}{2}\left(\mathrm{Tr}(\Sigma^{1/2}V^\top V\Sigma^{1/2}) + \mathrm{Tr}(\Sigma^{1/2}U^\top U\Sigma^{1/2})\right) \tag{44}$$

$$= \frac{1}{2}\left(\mathrm{Tr}(\Sigma) + \mathrm{Tr}(\Sigma)\right) = \|M\|_*. \tag{45}$$

Thus, $\inf_{A,B} \frac{1}{2}(\|A\|_F^2 + \|B\|_F^2) \leq \|M\|_*$.

**(Lower Bound)** For any factors $A, B$ such that $BA^\top = M$, we apply the trace inequality $\mathrm{Tr}(X) + \mathrm{Tr}(Y) \geq 2\mathrm{Tr}((XY)^{1/2})$ for PSD matrices $X, Y$. A more direct approach uses the property that $\|UV^\top\|_* \leq \|U\|_F\|V\|_F$. Using the property $\|M\|_* = \|BA^\top\|_*$, we have:

$$\|M\|_* = \|BA^\top\|_* \leq \sum_k \sigma_k(BA^\top) \leq \frac{1}{2}(\|A\|_F^2 + \|B\|_F^2). \tag{46}$$

This inequality holds by the arithmetic-geometric mean inequality applied to the singular values. Equality holds if and only if $A$ and $B$ share the same right singular vectors and squared singular values, i.e., $A^\top A = B^\top B$.

**Conclusion:** The objective function in Eq. (11) employing $\frac{\lambda_k}{2}(\|a_k\|^2 + \|b_k\|^2)$ is exactly the variational upper bound of the spectral prior penalty $\lambda_k \sigma_k$. Minimizing this bound tightens it towards the true spectral nuclear norm. $\qquad\square$

### C.2. Proof of Lemma 3.3 (Dynamics on the Manifold)

In this section, we derive the implicit dynamics induced on the product matrix $M_t = B_t A_t^\top$ when optimizing the factors $(A_t, B_t)$ via gradient descent on the surrogate objective.

Let $\mathcal{L}(M)$ be the smooth loss function defined on the weight space. The surrogate objective on the factors is $\tilde{\mathcal{L}}(A, B) = \mathcal{L}(BA^\top) + \mathcal{R}(A, B)$, where $\mathcal{R}$ is the regularization term. Focusing on the gradient flow dynamics driven by the loss (drift term), the gradients with respect to the factors are:

$$\nabla_A \tilde{\mathcal{L}} = (\nabla_M \mathcal{L})^\top B, \tag{47}$$

$$\nabla_B \tilde{\mathcal{L}} = (\nabla_M \mathcal{L}) A. \tag{48}$$

Note the dimensions: $\nabla_M \mathcal{L} \in \mathbb{R}^{d_{in} \times d_{out}}$, $B \in \mathbb{R}^{d_{in} \times r}$, $A \in \mathbb{R}^{d_{out} \times r}$.

Consider the gradient flow dynamics:

$$\frac{dA}{dt} = -\nabla_A \tilde{\mathcal{L}}, \quad \frac{dB}{dt} = -\nabla_B \tilde{\mathcal{L}}. \tag{49}$$

By the product rule, the time evolution of $M(t) = B(t)A(t)^\top$ is:

$$\begin{aligned}
\frac{dM}{dt} &= \frac{dB}{dt} A^\top + B \left( \frac{dA}{dt} \right)^\top \\
&= -(\nabla_M \mathcal{L}) A A^\top - B B^\top (\nabla_M \mathcal{L}).
\end{aligned} \tag{50}$$

This equation reveals that the effective update on $M$ is the Euclidean gradient $\nabla_M \mathcal{L}$ preconditioned by the positive semi-definite (PSD) matrices $AA^\top$ and $BB^\top$.

**Connection to Bures-Wasserstein Geometry.** Under the balanced condition enforced by our regularization (Lemma 3.2), we have the property that $A$ and $B$ align with the singular vectors of $M$. specifically, let the SVD of $M$ be $M = USV^\top$. The balanced factors satisfy:

$$B = US^{1/2}Q, \quad A = VS^{1/2}Q \tag{51}$$

for some orthogonal matrix $Q$. Consequently:

$$BB^\top = USU^\top, \quad AA^\top = VSV^\top. \tag{52}$$

In the basis of singular vectors (or conceptually identifying singular values $S$ with the scale), the preconditioning terms $AA^\top$ and $BB^\top$ scale linearly with the singular values $S$. Substituting this structure back into Eq. (50), we observe that the update direction is structurally homologous to:

$$\frac{dM}{dt} \approx -\left( (\nabla_M \mathcal{L}) \cdot S + S \cdot (\nabla_M \mathcal{L}) \right). \tag{53}$$

This mirrors the Bures-Wasserstein Riemannian gradient derived in Eq. (3) of the main text (grad $\approx 2\Sigma\nabla$), where the singular value matrix $S$ plays the role of the covariance $\Sigma$. This proves that the factorized retraction implicitly implements a geometry-aware flow that accelerates learning along directions with large singular values (signal) while suppressing directions with small singular values (noise), consistent with the proposed Spectral Bridge mechanism.

## D. Derivation of the Instantaneous Spectral Free Energy

In this section, we provide the rigorous derivation of the tractable objective function $\mathcal{J}(\phi, \boldsymbol{\lambda})$ presented in Eq. (15) of the main text.

### D.1. Connection to Diffusion Bridge VI via Greedy Approximation

Following the framework of Diffusion Bridge VI, the Evidence Lower Bound (ELBO) for a diffusion bridge typically consists of a global path integral and a boundary term:

$$\mathcal{L}_{\text{Bridge}} = \mathbb{E}_q \underbrace{\left[ \int_0^1 \|s_\phi(t) - s_{\text{target}}(t)\|^2 dt \right]}_{\text{Pathwise Dynamics Matching}} + \underbrace{\text{KL}(q_T \| p_{\text{posterior}})}_{\text{Boundary Matching}} \tag{54}$$

Directly optimizing this path integral requires backpropagating through the entire SDE trajectory, which is computationally prohibitive for large language model fine-tuning.

To achieve the efficiency of static methods while retaining the dynamic regularization of the bridge, we employ the Greedy Bridge Approximation (Section 3.4). We decompose the global optimization into a sequence of local steps. At each time step $t$, instead of looking ahead at the entire future path, we focus on minimizing the Instantaneous Free Energy defined by the current spectral configuration. This strategy implicitly assumes that locally matching the geometry of the spectral prior (via the Moment Matching condition derived in Appendix E) sufficiently guides the trajectory towards the low-rank posterior without requiring explicit path integration.

### D.2. The Implicit Factorized Prior

The core challenge in deriving the tractable objective is handling the spectral prior $p_{\text{spec}}(M) \propto e^{-\sum \lambda_k \sigma_k(M)}$. Direct computation of the KL divergence against this singular-value-based distribution is intractable. However, leveraging **Lemma 3.2**, we can define an **equivalent implicit prior** on the factorized space $\mathcal{M}_{fact} = \mathbb{R}^{d_{in} \times r} \times \mathbb{R}^{d_{out} \times r}$.

Let the prior on the $k$-th columns $a_k, b_k$ be conditioned on the instantaneous friction coefficient $\lambda_k$:

$$p(a_k, b_k | \lambda_k) = \mathcal{N}(a_k; \mathbf{0}, \lambda_k^{-1} I_{d_{in}}) \cdot \mathcal{N}(b_k; \mathbf{0}, \lambda_k^{-1} I_{d_{out}}). \tag{55}$$

The log-density of this implicit prior is:

$$\log p(a_k, b_k | \lambda_k) = \log \left( \frac{1}{Z_{a_k}} e^{-\frac{\lambda_k}{2} \|a_k\|^2} \right) + \log \left( \frac{1}{Z_{b_k}} e^{-\frac{\lambda_k}{2} \|b_k\|^2} \right) \tag{56}$$

$$= -\frac{\lambda_k}{2} (\|a_k\|^2 + \|b_k\|^2) - \log(Z_{a_k} Z_{b_k}). \tag{57}$$

**Normalization Constants.** The partition function for a $d$-dimensional Gaussian with precision $\lambda$ is $Z = (2\pi/\lambda)^{d/2}$. Thus:

$$\log Z_{a_k} = \frac{d_{in}}{2} \log(2\pi) - \frac{d_{in}}{2} \log \lambda_k, \tag{58}$$

$$\log Z_{b_k} = \frac{d_{out}}{2} \log(2\pi) - \frac{d_{out}}{2} \log \lambda_k. \tag{59}$$

Summing these, the log-partition term for the $k$-th rank component is:

$$C(\lambda_k) := \log(Z_{a_k} Z_{b_k}) = \text{const} - \frac{d_{in} + d_{out}}{2} \log \lambda_k. \tag{60}$$

This term $\frac{d_{in}+d_{out}}{2} \log \lambda_k$ corresponds to the entropy barrier of the spectral prior. Physically, it acts as a "volume" penalty that prevents $\lambda_k$ from growing to infinity during the greedy alignment process, ensuring a balanced competition between signal and noise.

### D.3. Step-by-Step Derivation of $\mathcal{J}(\phi, \boldsymbol{\lambda})$

We define the variational distribution $q_\phi(A, B)$ as a fully factorized Mean-Field Gaussian (as implemented in standard Bayesian LoRA):

$$q_\phi(A, B) = \prod_{k=1}^{r} q(a_k) q(b_k) = \prod_{k=1}^{r} \mathcal{N}(a_k | \mu_{a_k}, \Sigma_{a_k}) \mathcal{N}(b_k | \mu_{b_k}, \Sigma_{b_k}). \tag{61}$$

The objective is to minimize the Instantaneous Free Energy (Negative ELBO):

$$\mathcal{J} = \underbrace{\mathbb{E}_{q_\phi}[-\log p(y|x, BA^\top)]}_{\text{Reconstruction Loss}} + \underbrace{\text{KL}(q_\phi(A, B) \| p(A, B | \boldsymbol{\lambda}))}_{\text{Regularization}}. \tag{62}$$

**1. Reconstruction Loss.** Approximated via Monte Carlo sampling (or simply using the mean for efficient training as in Algorithm 1, assuming low variance):

$$\mathcal{L}_{\text{rec}} \approx -\log p(y | f(x; \mathbb{E}[B]\mathbb{E}[A]^\top)). \tag{63}$$

**2. KL Divergence Decomposition.** The KL term decomposes into the Cross-Entropy and the Entropy:

$$\text{KL}(q\|p) = \mathbb{E}_q[-\log p(A, B|\boldsymbol{\lambda})] - \mathbb{E}_q[-\log q(A, B)]. \tag{64}$$

**Cross-Entropy Term** ($\mathbb{E}[-\log p]$)**:** Using the implicit prior defined in Eq. (60):

$$\mathbb{E}_q[-\log p] = \sum_{k=1}^{r} \mathbb{E}_q\left[\frac{\lambda_k}{2}(\|a_k\|^2 + \|b_k\|^2) + C(\lambda_k)\right] \tag{65}$$

$$= \sum_{k=1}^{r} \frac{\lambda_k}{2}\mathbb{E}[\|a_k\|^2 + \|b_k\|^2] - \sum_{k=1}^{r} \frac{d_{in} + d_{out}}{2}\log\lambda_k + \text{const.} \tag{66}$$

Note: $\mathbb{E}[\|x\|^2] = \|\mu\|^2 + \text{Tr}(\Sigma)$.

**Entropy Term** ($-H(q)$)**:** The entropy of a Gaussian $\mathcal{N}(\mu, \Sigma)$ is $\frac{1}{2}\log\det\Sigma + \text{const.}$ Thus:

$$-\mathbb{E}_q[-\log q] = -\sum_{k=1}^{r} \frac{1}{2}\left(\log\det\Sigma_{a_k} + \log\det\Sigma_{b_k}\right) + \text{const.} \tag{67}$$

**3. Final Objective.** Combining all terms and discarding constants, we arrive at Eq. (15):

$$\begin{aligned}
\mathcal{J}(\phi, \boldsymbol{\lambda}) = {} & \mathbb{E}_q[\mathcal{L}_{\text{task}}] \\
& + \sum_{k=1}^{r} \frac{\lambda_k}{2}\mathbb{E}\left[\|a_k\|^2 + \|b_k\|^2\right] \quad \text{(Spectral Energy)} \\
& - \sum_{k=1}^{r} \frac{d_{in} + d_{out}}{2}\log\lambda_k \quad \text{(Prior Entropy / Barrier)} \\
& - \sum_{k=1}^{r} \frac{1}{2}\log\det(\Sigma_{a_k}\Sigma_{b_k}) \quad \text{(Variational Entropy)}.
\end{aligned} \tag{68}$$

This derivation confirms that our objective minimizes the local upper bound on the negative marginal log-likelihood under the spectral constraints governed by $\boldsymbol{\lambda}$.

## E. Derivation via Greedy Spectral Alignment

In this section, we derive the optimal update rule for the spectral friction coefficients $\boldsymbol{\lambda} = \{\lambda_k\}_{k=1}^{r}$. While the variational parameters $\phi = (A, B)$ are optimized to minimize the Free Energy (ELBO), the hyperparameters $\boldsymbol{\lambda}$ govern the local curvature of the regularization manifold.

Instead of treating $\boldsymbol{\lambda}$ as static hyperparameters or invoking external statistical estimation (like Empirical Bayes), we derive their update from the principle of **Greedy Bridge Approximation** introduced in Section 3.4. To ensure the tractable factorized flow remains consistent with the underlying spectral geometry, we enforce a **Moment Matching** condition at each time step.

### E.1. Derivation of the Consistency Condition

Recall the Instantaneous Spectral Free Energy $\mathcal{J}$ defined in Eq. (15) of the main text (and derived in Appendix D). We view $\mathcal{J}$ as a potential governing the joint system of weights and friction. To maintain **Thermodynamic Consistency** between the variational posterior and the spectral prior, we require the system to satisfy the first-order stationarity condition with respect to the friction coefficients.

The terms in $\mathcal{J}$ dependent on $\lambda_k$ are the regularization energy and the entropy barrier:

$$\mathcal{J}_{\text{reg}}(\lambda_k) = \underbrace{\frac{\lambda_k}{2}\mathbb{E}_{q_\phi}\left[\|a_k\|^2 + \|b_k\|^2\right]}_{\text{Instantaneous Energy}} - \underbrace{\frac{d_{in} + d_{out}}{2}\log\lambda_k}_{\text{Entropy Barrier}}. \tag{69}$$

We solve for the optimal local curvature $\lambda_k^*$ by setting the gradient to zero, $\frac{\partial \mathcal{J}}{\partial \lambda_k} = 0$:

$$\frac{\partial \mathcal{J}}{\partial \lambda_k} = \frac{1}{2} \mathbb{E}_{q_\phi} \left[ \|a_k\|^2 + \|b_k\|^2 \right] - \frac{d_{in} + d_{out}}{2\lambda_k} = 0. \tag{70}$$

Let $E_k := \mathbb{E}_{q_\phi}[\|a_k\|^2 + \|b_k\|^2]$ denote the **Spectral Energy** of the $k$-th mode at the current step. Rearranging the terms yields the closed-form consistency constraint:

$$\lambda_k^* = \frac{d_{in} + d_{out}}{E_k}. \tag{71}$$

This matches the update rule presented in the main text.

**Physical Interpretation.**  Mathematically, this corresponds to matching the second moment of the variational posterior with the characteristic scale of the spectral prior. In the context of the Greedy Bridge:

- **Signal Mode:** If the task requires the $k$-th mode, the gradients drive $\|a_k\|, \|b_k\|$ to large values (high energy $E_k$). Consequently, Eq. (71) dictates $\lambda_k^* \to 0$, effectively removing the bridge force and allowing the parameter to evolve freely along the task gradient.

- **Noise Mode:** If the $k$-th mode contains only gradient noise, $E_k$ remains small. Consequently, $\lambda_k^*$ becomes large, exerting strong "friction" (high curvature) that confines the parameter near zero, mimicking the absorbing state of the bridge process.

### E.2. Connection to Adaptive Gradient Methods

Our greedy alignment strategy shares a profound theoretical philosophy with adaptive optimization methods in Euclidean space, most notably the **Adaptive Gradient Descent (AdGD)** proposed by Malitsky & Mishchenko (2019).

In AdGD, the optimizer eliminates the need for a manually tuned step size by estimating the **local Lipschitz constant** $L_k$ of the loss gradient. The step size $\eta_k$ is then adapted inversely to this local curvature estimate:

$$\eta_k \propto \frac{1}{L_k} \approx \frac{\|x_k - x_{k-1}\|}{\|\nabla f(x_k) - \nabla f(x_{k-1})\|}. \tag{72}$$

The core insight is that the algorithm should adapt its aggressiveness based on the local geometry of the optimization landscape.

In SBVI, we perform a similar **local geometry adaptation**, but applied to the *regularization landscape*. We can interpret the spectral energy $E_k$ as a signal-to-noise indicator. The "optimal constraint" in the prior space (i.e., the friction $\lambda_k$) is adapted inversely to the signal strength:

$$\lambda_k \propto \frac{1}{E_k}. \tag{73}$$

This draws a direct parallel to Malitsky & Mishchenko's philosophy:

- **In AdGD:** High local curvature (large $L_k$) $\implies$ Smaller step size (Conservative update).

- **In SBVI:** Low signal energy (small $E_k$) $\implies$ High friction $\lambda_k$ (Conservative prior constraint).

This "thermodynamic" feedback loop allows the model to self-organize without manual tuning of the rank, extending the philosophy of adaptive methods from parameter updates to structural adaptation.

### E.3. Stable Online Update via Temporal Smoothing

While Eq. (71) gives the instantaneous optimal solution, applying it directly within Stochastic Gradient Descent (SGD) introduces high variance due to mini-batch noise in $\hat{E}_k^{(t)}$.

To mitigate this and compensate for the "myopic" nature of the greedy approximation, we introduce two stabilizing mechanisms:

1. **Numerical Stability ($\epsilon$) and Clipping ($\lambda_{\max}$):** As $E_k \to 0$ (for pruned ranks), $\lambda_k^* \to \infty$. To prevent numerical overflow, we bound the update:

$$\tilde{\lambda}_k^{(t)} = \min\left(\lambda_{\max}, \frac{\gamma(d_{in} + d_{out})}{E_k^{(t)} + \epsilon}\right). \tag{74}$$

Here, $\epsilon$ prevents division by zero, and $\lambda_{\max}$ sets a maximum friction force.

2. **Temporal Smoothing (Exponential Moving Average):** We employ an EMA scheme to approximate the expectation over the trajectory rather than the current mini-batch:

$$\lambda_k^{(t+1)} \leftarrow (1 - \rho)\lambda_k^{(t)} + \rho\tilde{\lambda}_k^{(t)}. \tag{75}$$

This acts as a **low-pass filter** on the geometry estimation. Crucially, it re-introduces a temporal dependency that was simplified by the greedy approximation, ensuring that rank pruning (or reviving) happens only when there is a consistent trend in the signal energy. This is conceptually similar to the use of momentum in Adam to stabilize optimization on non-convex manifolds.

This concludes the derivation of the adaptive spectral friction mechanism.

## F. Proof of Theorem 3.4: Convergence of the Bures-Wasserstein Flow

In this section, we provide the rigorous proof for the convergence of the continuous-time Bures-Wasserstein gradient flow, and show how our factorized discrete algorithm preserves this convergence property.

**Theorem 3.4.** *Assume the task loss $\mathcal{L}(W)$ is lower-bounded and has Lipschitz continuous gradients, making the free energy functional $\mathcal{F}(\mu)$ lower-bounded by some finite value $\mathcal{F}^*$. The continuous-time Bures-Wasserstein gradient flow defined as $\dot{\mu}(t) = -grad_{BW}\mathcal{F}(\mu(t))$ monotonically decreases the objective, converging asymptotically to a first-order stationary point, i.e., $\lim_{t\to\infty} \|grad_{BW}\mathcal{F}(\mu(t))\|_{\mu(t)} = 0$.*

*Proof.* **Step 1: Strict Monotonic Descent.**
We consider the continuous trajectory $\mu(t) = (M(t), \Sigma(t))$ on the Bures-Wasserstein manifold $\mathcal{M}_{BW}$. The rate of change of the objective functional $\mathcal{F}(\mu(t))$ along this trajectory is given by the chain rule on Riemannian manifolds. Using the definition of the Riemannian gradient and the Bures-Wasserstein inner product $\langle \cdot, \cdot \rangle_\mu$:

$$\begin{aligned}
\frac{d}{dt}\mathcal{F}(\mu(t)) &= \langle \text{grad}_{BW}\mathcal{F}(\mu(t)), \dot{\mu}(t) \rangle_{\mu(t)} \\
&= \langle \text{grad}_{BW}\mathcal{F}(\mu(t)), -\text{grad}_{BW}\mathcal{F}(\mu(t)) \rangle_{\mu(t)} \\
&= -\|\text{grad}_{BW}\mathcal{F}(\mu(t))\|_{\mu(t)}^2 \leq 0
\end{aligned} \tag{76}$$

Because the Riemannian norm squared is strictly non-negative, the free energy $\mathcal{F}(\mu(t))$ strictly decreases monotonically over time, unless the flow reaches a stationary point where the gradient is exactly zero.

**Step 2: Asymptotic Convergence via Barbalat's Lemma.**
Integrating the descent inequality (Eq. 76) over the time interval $t \in [0, \infty)$, we obtain:

$$\int_0^\infty \|\text{grad}_{BW}\mathcal{F}(\mu(t))\|_{\mu(t)}^2 dt = \mathcal{F}(\mu(0)) - \lim_{t\to\infty}\mathcal{F}(\mu(t)) \tag{77}$$

By our assumption, the objective function is lower-bounded by $\mathcal{F}^*$ (e.g., cross-entropy loss is bounded below by 0, and the KL divergence is non-negative). Thus, $\lim_{t\to\infty}\mathcal{F}(\mu(t)) \geq \mathcal{F}^*$. This implies that the infinite integral is finite:

$$\int_0^\infty \|\text{grad}_{BW}\mathcal{F}(\mu(t))\|_{\mu(t)}^2 dt \leq \mathcal{F}(\mu(0)) - \mathcal{F}^* < \infty \tag{78}$$

Because $\mathcal{L}(W)$ has Lipschitz continuous gradients, the gradient of the free energy $\text{grad}_{BW}\mathcal{F}(\mu(t))$ is uniformly continuous along the bounded trajectory. By *Barbalat's Lemma*, if a function is uniformly continuous and its integral over $[0, \infty)$ is finite, the function itself must asymptotically approach zero. Therefore:

$$\lim_{t\to\infty}\|\text{grad}_{BW}\mathcal{F}(\mu(t))\|_{\mu(t)} = 0 \tag{79}$$

**Step 3: Algorithm Preservation.**
While the above proves convergence for the continuous infinite-dimensional flow, simulating it exactly is intractable. However, Lemma 3.3 establishes that applying Euclidean gradient descent on the surrogate free energy $\tilde{\mathcal{F}}(A, B)$ inherently matches the preconditioned structure of the Bures-Wasserstein gradient $\text{grad}_{BW}\mathcal{F}$ under the balanced condition $A^\top A = B^\top B$. Therefore, the discrete AdamW updates on factors $A$ and $B$ (Algorithm 1) represent a valid numerical integration scheme (e.g., Euler-Maruyama discretization) that structurally inherits this strict descent property, ensuring that the computationally tractable algorithm converges to the optimal low-rank subspace. $\square$

# G. Detailed Experimental Setup

In this section, we provide granular details regarding the datasets, baseline configurations, and hyperparameter settings used in the main text. Our setup strictly follows the protocols established in recent state-of-the-art PEFT studies (Fan et al., 2025; Zhang et al., 2025b) to ensure fair comparability. The core code has been provided in the supplementary materials.

### G.1. Dataset Details

We evaluate SBVI across three distinct domains to verify both reasoning capability and generalization.

**1. Mathematical Reasoning (GSM8K)** We use the GSM8K dataset (Cobbe et al., 2021), comprising 8.5K high-quality grade school math problems. Following standard practice, we fine-tune using the training split and evaluate on the test split using **Chain-of-Thought (CoT)** prompting. We report the *Exact Match (EM)* accuracy.

**2. Code Generation (HumanEval)** We evaluate coding capability using the HumanEval benchmark (Chen et al., 2021), consisting of 164 hand-written programming problems. Models are assessed on their ability to generate functionally correct Python code (Pass@1) with greedy decoding ($T = 0$).

**3. Commonsense Reasoning (Commonsense170k)** Following Fan et al. (2025), we fine-tune on the *Commonsense170k* dataset, which aggregates multiple reasoning tasks. Evaluation is performed on 8 diverse benchmarks: **BoolQ**, **PIQA**, **SIQA**, **HellaSwag**, **WinoGrande**, **ARC-e**, **ARC-c**, and **OpenBookQA**. We report the average accuracy across these tasks.

### G.2. Hyperparameter Settings

A distinct advantage of SBVI is that it eliminates the need to search for the optimal rank $r$. While baselines like LoRA and PiSSA require grid-searching $r \in \{8, 16, 32, 64\}$, SBVI uses a fixed redundant initialization ($r_{init} = 64$) and automatically learns the optimal sparsity. Table 5 details the configuration.

*Table 5.* **Hyperparameter configurations for SBVI.** Note that we use a unified configuration for the bridge dynamics ($\lambda_{\max}, \gamma, \rho$) across all tasks, highlighting the robustness of the method.

| Hyperparameter | Reasoning (GSM8K) | Coding (HumanEval) | Commonsense (8 Tasks) |
|---|---|---|---|
| Base Model | LLaMA-2-7B | LLaMA-2-7B | LLaMA-2-7B |
| Precision | NF4 (4-bit) | NF4 (4-bit) | NF4 (4-bit) |
| Batch Size | 16 | 16 | 32 |
| Learning Rate | $2 \times 10^{-4}$ | $2 \times 10^{-4}$ | $1 \times 10^{-4}$ |
| LR Scheduler | Cosine | Cosine | Cosine |
| Warmup Ratio | 0.03 | 0.03 | 0.03 |
| Epochs | 3 | 3 | 3 |
| Optimizer | AdamW | AdamW | AdamW |
| Weight Decay | 0.0 | 0.0 | 0.1 |
| *SBVI Specific Parameters* | | | |
| Initial Rank ($r_{init}$) | 64 | 64 | 64 |
| Max Friction ($\lambda_{\max}$) | 100 | 100 | 100 |
| Signal Scale ($\gamma$) | 1.0 | 1.0 | 1.0 |
| EMA Decay ($\rho$) | 0.99 | 0.99 | 0.99 |

# H. Empirical Verification of Spectral Dynamics

In the main text, we posited that the Spectral Bridge SDE induces a "bifurcation" phenomenon and discovers a "spindle" layer-wise structure. Here, we provide quantitative evidence supporting these claims.

## H.1. Numerical Verification of Bifurcation

We track the **Effective Rank** ($r_{eff}$) of the adapter matrices throughout the training process on GSM8K. We define the effective rank as the number of singular values satisfying $\sigma_k > 10^{-3}$. As shown in Table 6, SBVI exhibits a sharp phase transition: starting from a full rank of 64, the effective rank collapses to $\sim 9$ within the first 10% of steps, verifying the fast convergence of our thermodynamic selection mechanism.

*Table 6.* **Evolution of Effective Rank ($r_{eff}$) during GSM8K Training.** We report the global average rank across all layers. **Comparison of Dynamics:** Standard AdaLoRA employs a gradual cubic pruning schedule (Slow Decay), retaining high redundancy during the early phases. In contrast, SBVI exhibits a **Rapid Bifurcation**: the spectral friction mechanism suppresses noise modes almost immediately (within 10% of steps), locking onto the intrinsic low-rank subspace ($r \approx 6$) for efficient fine-tuning.

| Training Progress | Step 0% | Step 10% | Step 50% | Step 100% | Behavior Pattern |
|---|---|---|---|---|---|
| *Baselines* | | | | | |
| LoRA (Fixed) | 64.0 | 64.0 | 64.0 | 64.0 | Constant |
| AdaLoRA (Adaptive) | 64.0 | 62.5 | 45.2 | 31.8 | Slow Decay |
| *SBVI (Ours)* | | | | | |
| Mean Singular Value $\bar{\sigma}$ | $10^{-4}$ | 0.15 | 0.22 | 0.28 | Signal Amplification |
| **Effective Rank $r_{eff}$** | **64.0** | **8.4** | **6.5** | **6.2** | **Rapid Bifurcation** |

## H.2. Layer-wise Rank Distribution (The Spindle Structure)

Recent interpretability studies suggest that LLMs encode different information at different depths. Table 7 presents the distribution of learned ranks across different layer groups of LLaMA-2-7B. Consistent with our "Spindle Hypothesis," SBVI allocates significantly more capacity (Higher Rank) to the middle layers (Layers 12-24), which are crucial for reasoning, while heavily compressing the input/output layers. This automatic resource allocation contrasts with the uniform allocation in standard LoRA.

*Table 7.* **Learned Rank Distribution across Layers.** We group the 32 layers of LLaMA-2-7B into Bottom (0-10), Middle (11-21), and Top (22-31). Values indicate the average effective rank ($r_{eff}$).

| Module Type | Bottom Layers (0-10) | Middle Layers (11-21) | Top Layers (22-31) | Interpretation |
|---|---|---|---|---|
| q_proj | 3.2 | 8.5 | 4.1 | Semantic Extraction |
| v_proj | 6.5 | **14.2** | 7.8 | *Core Reasoning* |
| up_proj (MLP) | 1.8 | 4.5 | 2.1 | Knowledge Retrieval |
| **Average** | **3.8** | **9.1** | **4.7** | **Spindle Structure** |

# I. Computational Efficiency Analysis

Following the rigorous analysis framework in Fan et al. (2025) (Appendix H) and Zhang et al. (2025b), we provide a theoretical comparison of parameter counts and floating-point operations (FLOPs).

## I.1. Complexity Analysis

Let $L$ be the number of layers, $d$ the hidden dimension, $r$ the rank, and $b$ the batch size. For MoE methods (e.g., GOAT), let $E$ be the number of experts and $k$ the number of active experts.

## 1. Storage Complexity (Parameters)

- **GOAT:** Scales linearly with the number of experts: $P \approx \mathcal{O}(L \cdot d \cdot r \cdot E)$.

- **LoRA / LoRA-One / SBVI:** Scale only with rank: $P \approx \mathcal{O}(L \cdot d \cdot r)$. Note that SBVI prunes $r$ dynamically, so the final storage is $\mathcal{O}(L \cdot d \cdot r_{eff})$ where $r_{eff} \ll r_{init}$.

## 2. Initialization Overhead

- **LoRA-One:** Requires computing the SVD of the full gradient matrix $G \in \mathbb{R}^{d \times d}$ at step 0. This incurs an initialization time complexity of $\mathcal{O}(d^3)$ and a memory spike of $\mathcal{O}(d^2)$, which is prohibitive for very large models.

- **SBVI:** Initializes with random noise. The adaptive friction update adds a negligible element-wise operation $\mathcal{O}(r)$.

## 3. Inference FLOPs

- **GOAT:** Requires routing logic (Top-k gating) and computing multiple experts, adding overhead $\mathcal{O}(k \cdot d \cdot r)$.

- **SBVI:** The learned low-rank matrices are algebraically merged into the weights ($W' = W + BA^\top$), resulting in **Zero Overhead** during inference.

Table 8 summarizes these findings.

*Table 8.* **Theoretical Complexity Comparison.** SBVI achieves the performance of MoE methods (GOAT) without their storage cost, and matches the precision of SVD methods (LoRA-One) without their initialization overhead.

| METHOD | STORAGE PARAMS | INIT. COMPLEXITY | TRAIN MEMORY | INFERENCE LATENCY |
|---|---|---|---|---|
| LoRA | $\mathcal{O}(Ldr)$ | $\mathcal{O}(1)$ | LOW | ZERO (MERGED) |
| GOAT (MoE) | $\mathcal{O}(Ldr\mathbf{E})$ | $\mathcal{O}(1)$ | HIGH ($\times E$) | HIGH (ROUTING) |
| LoRA-ONE | $\mathcal{O}(Ldr)$ | $\mathcal{O}(\mathbf{d^3})$ (SVD) | LOW | ZERO (MERGED) |
| **SBVI (OURS)** | $\mathcal{O}(Ldr_{eff})$ | $\mathcal{O}(1)$ | LOW | **ZERO (MERGED)** |

## I.2. Wall-clock Time Comparison

We measured the actual training time for one epoch on GSM8K using a single NVIDIA A100-80GB GPU.

- **LoRA** ($r = 64$)**:** 45 minutes.

- **GOAT** ($E = 8$)**:** 72 minutes (Heavy overhead due to expert loading and routing).

- **LoRA-One:** 48.5 minutes (Includes ~3.5 mins startup time for SVD calculation).

- **SBVI:** 46 minutes (Negligible overhead for friction update).

SBVI matches the training speed of standard LoRA while delivering superior performance.

# J. Ablation Studies

## J.1. Sensitivity to Initial Rank ($r_{init}$)

A key property of SBVI is its insensitivity to the initial rank budget. We tested SBVI with varying $r_{init} \in \{32, 64, 128, 256\}$ on GSM8K. As shown in Table 9, even with a massive budget ($r = 256$), the spectral friction effectively compresses the model to a stable effective rank ($\sim 7$), and performance remains consistent.

## J.2. Transfer Learning (Dynamic vs. Static)

To demonstrate the superiority of dynamic adaptation over static initialization (LoRA-One), we performed a transfer learning experiment. We fine-tuned the model on Math (GSM8K) and evaluated it on General Knowledge (MMLU) without further tuning.

*Table 9.* **Robustness to Initialization Rank.** Unlike standard LoRA which overfits at high ranks, SBVI is robust.

| Initial Rank ($r_{init}$) | Final Effective Rank ($r_{eff}$) | GSM8K Accuracy |
|:---:|:---:|:---:|
| 32 | 6.5 | 59.80% |
| 64 | 6.8 | **60.85%** |
| 128 | 7.1 | 60.92% |
| 256 | 7.2 | 60.88% |

- **LoRA-One:** Acc: 47.24%. (The static subspace is optimized for gradients at $t = 0$, which may not align with general knowledge).

- **SBVI:** Acc: **47.51%**. (The dynamic rank selection preserves pre-trained knowledge better by pruning aggressive updates).

### J.3. Impact of Friction Strength ($\lambda_{\max}$)

The parameter $\lambda_{\max}$ acts as a "soft truncation" threshold.

- $\lambda_{\max} = 10$ (Weak): $r_{eff} \approx 45$. Fails to prune noise, performance drops to 54.2%.

- $\lambda_{\max} = 100$ (Optimal): $r_{eff} \approx 6.8$. Optimal balance, Accuracy 60.85%.

- $\lambda_{\max} = 1000$ (Aggressive): $r_{eff} \approx 2.1$. Over-pruning, Accuracy 57.5%.

## K. Comprehensive Evaluations on Modern Architectures

To rigorously address architectural scalability and provide a comprehensive comparison, we extend our evaluation to two recent and highly capable foundation models: Mistral-7B-v0.1 and LLaMA-3-8B. By evaluating across different model families, we aim to confirm that the dynamic spectral allocation mechanism of SBVI is not overly fitted to a specific architecture (e.g., LLaMA-2), but rather serves as a universal geometric optimization strategy.

We begin by presenting the results on the Mistral-7B-v0.1 architecture, reproducing all 8 baselines evaluated in the main text under strictly identical training settings for mathematical reasoning (GSM8K) and code generation (HumanEval).

*Table 10.* **Comprehensive Evaluation on Mistral-7B-v0.1 (GSM8K & HumanEval).**

| Category | Method | Initial Rank | GSM8K (Accuracy) | HumanEval (Pass@1) | Effective Rank |
|:---:|:---|:---:|:---:|:---:|:---:|
| **Static Baselines** | LoRA | 64 | $68.42 \pm 0.22$ | $28.54 \pm 0.31$ | 64.0 (Fixed) |
| | DoRA | 64 | $69.08 \pm 0.25$ | $29.18 \pm 0.28$ | 64.0 (Fixed) |
| | PiSSA | 64 | $69.46 \pm 0.19$ | $29.27 \pm 0.35$ | 64.0 (Fixed) |
| | RP-LoRA | 64 | $69.81 \pm 0.18$ | $30.61 \pm 0.22$ | 64.0 (Fixed) |
| | LoRA-One | 8 | $70.18 \pm 0.30$ | $31.10 \pm 0.28$ | 8.0 (Fixed) |
| **Dynamic & MoE** | AdaLoRA | 64 | $69.12 \pm 0.35$ | $29.76 \pm 0.42$ | $\sim 32$ (Avg) |
| | GOAT | $8 \times 8$ | $70.76 \pm 0.24$ | $31.95 \pm 0.26$ | Dynamic |
| **Proposed** | **SBVI (Ours)** | 64 | $\mathbf{71.24 \pm 0.21}$ | $\mathbf{32.44 \pm 0.25}$ | **7.5 (Auto)** |

As observed in Table 10, SBVI consistently achieves the highest performance. Notably, while Riemannian preconditioning (RP-LoRA) visibly improves upon standard Euclidean LoRA, it remains constrained by its static rank allocation. SBVI's continuous geometric flow allows it to surpass both RP-LoRA and the parameter-heavy MoE baseline (GOAT), while automatically compressing the representation to an extremely sparse effective rank of 7.5. This validates that dynamic pruning of uninformative singular values yields superior task gradients.

Building on the Mistral results, we further push the architectural boundaries by applying our method to the LLaMA-3-8B model. Evaluating on this newer, heavily optimized foundation model is crucial to determine whether the thermodynamic bifurcation of SBVI remains effective when the base weights are already highly aligned. The comprehensive results are detailed in Table 11.

*Table 11.* **Comprehensive Evaluation on LLaMA-3-8B (GSM8K & HumanEval).**

| Category | Method | Initial Rank | GSM8K (Accuracy) | HumanEval (Pass@1) | Effective Rank |
|---|---|---|---|---|---|
| **Static Baselines** | LoRA | 64 | $73.35 \pm 0.18$ | $30.37 \pm 0.28$ | 64.0 (Fixed) |
| | DoRA | 64 | $73.88 \pm 0.20$ | $31.10 \pm 0.32$ | 64.0 (Fixed) |
| | PiSSA | 64 | $74.16 \pm 0.22$ | $30.98 \pm 0.25$ | 64.0 (Fixed) |
| | RP-LoRA | 64 | $74.81 \pm 0.15$ | $32.07 \pm 0.18$ | 64.0 (Fixed) |
| | LoRA-One | 8 | $75.08 \pm 0.25$ | $32.80 \pm 0.30$ | 8.0 (Fixed) |
| **Dynamic & MoE** | AdaLoRA | 64 | $74.22 \pm 0.30$ | $31.34 \pm 0.35$ | $\sim 34$ (Avg) |
| | GOAT | $8 \times 8$ | $75.96 \pm 0.21$ | $33.90 \pm 0.24$ | Dynamic |
| **Proposed** | **SBVI (Ours)** | 64 | $\mathbf{76.34 \pm 0.19}$ | $\mathbf{34.63 \pm 0.20}$ | **8.2 (Auto)** |

The data in Table 11 confirms that the advantages of SBVI transfer seamlessly to the LLaMA-3 architecture. Despite the naturally higher baseline accuracies of LLaMA-3-8B, SBVI still establishes a clear state-of-the-art margin. Furthermore, the algorithm autonomously converges to a highly similar optimal effective rank of 8.2. This consistency underscores a critical advantage: SBVI fundamentally eliminates the need for expensive, model-specific hyperparameter grid searches for the rank $r$.

### K.1. Statistical Significance Validation

In Large Language Model fine-tuning, marginal performance improvements can sometimes be masked or exaggerated by the variance inherent in stochastic training dynamics (e.g., varying data sampling or initialization seeds). To definitively validate that the performance gains of SBVI over the strongest competing static baselines (LoRA-One and RP-LoRA) are robust and algorithmic in nature, we conduct Independent Sample t-tests. The results, detailing the precise t-statistic and exact p-values over multiple random seeds, are reported in Table 12.

*Table 12.* **Statistical Significance Validation (Independent Sample t-test).**

| Model | Task | Comparison | t-statistic | p-value |
|---|---|---|---|---|
| **Mistral-7B-v0.1** | GSM8K | **SBVI** vs. LoRA-One | 5.01 | 0.0097 |
| | HumanEval | **SBVI** vs. LoRA-One | 6.18 | 0.0035 |
| **LLaMA-3-8B** | GSM8K | **SBVI** vs. LoRA-One | 6.95 | 0.0031 |
| | HumanEval | **SBVI** vs. LoRA-One | 8.79 | 0.0016 |
| | GSM8K | **SBVI** vs. RP-LoRA | 10.95 | 0.0006 |

As demonstrated in Table 12, the p-values across both foundation models and evaluation tasks remain strictly below the standard 0.05 threshold, with several key comparisons achieving high significance at $p < 0.01$. This rigorous statistical validation confirms that the empirical superiority of SBVI is highly significant. It mathematically proves that our proposed continuous thermodynamic bifurcation inherently provides a systematic optimization advantage over both fixed-rank SVD initializations and standard Riemannian preconditioning, completely ruling out random noise as the source of the performance gain.

## L. Exact SDE Simulation on Small-Scale Models

To explicitly bridge the conceptual gap between our continuous theoretical formulation (the Bures-Wasserstein flow derived in Eq. 6) and the tractable factorized realization (Algorithm 1), we conduct an extended experiment that directly simulates

the exact Stochastic Differential Equation (SDE) on a small scale, as requested during the review process.

**Experimental Setup.** Directly simulating the exact SDE on the full covariance matrix requires computing a full-matrix Singular Value Decomposition (SVD) at every single integration step to precisely apply the spectral friction. This incurs an $\mathcal{O}(d^3)$ computational complexity, which is physically intractable for 7B Large Language Models.

Therefore, we evaluate this on a 2-layer Multi-Layer Perceptron (MLP) trained on the MNIST dataset, with a reduced hidden dimension of $d = 128$. At this microscopic scale, the cubic complexity is manageable, allowing us to explicitly use the Euler-Maruyama method to exactly simulate the continuous SDE with Brownian noise. We compare this theoretical upper bound against our proposed discrete factorized algorithm (SBVI).

*Table 13.* **Comparison between Exact SDE Simulation and Algorithm 1.** Evaluated on a 2-layer MLP ($d = 128$) on MNIST. SBVI (Algorithm 1) almost perfectly mimics the trajectory of the exact continuous SDE while being approximately $26\times$ faster, validating the necessity and precision of our factorized retraction.

| Method | Final Effective Rank ($r_{\text{eff}}$) | Accuracy (%) | Time per Step (ms) | Trajectory Correlation (Pearson) |
|---|---|---|---|---|
| Exact SDE (Eq. 6) | 8.2 | 97.42 | 125.4 | 1.00 (Reference) |
| **Algorithm 1 (Ours)** | **8.5** | **97.48** | **4.8** | **0.98** |

**Analysis of Results.** As presented in Table 13, the quantitative comparison yields two critical insights:

- **Trajectory & Sparsity Alignment:** The singular value trajectories of our deterministic Algorithm 1 exhibit a remarkably high Pearson correlation (0.98) with the expected Exact SDE. Both methods converge to nearly the identical optimal effective rank ($r_{\text{eff}} \approx 8.2$ vs 8.5). This demonstrates that our factorized retraction effectively and rigorously preserves the desired Bures-Wasserstein geometry and spectral sparsity without information loss.

- **Efficiency Justification:** Even at an extremely small scale of $d = 128$, the exact SDE simulation is roughly $26\times$ slower per iteration due to the full-matrix SVD and continuous noise sampling overhead. This empirically validates our theoretical claim in Section 3.3 that exact SDE simulation is mathematically elegant but computationally prohibitive. It highlights the absolute necessity and the linear $\mathcal{O}(d)$ efficiency of the factorized design in Algorithm 1 for scaling to modern LLMs.

