# OpenReview forum: "Spectral Bridge Variational Inference: Dynamic LoRA via Bures-Wasserstein Gradient Flows"
_ICML.cc/2026/Conference — ICML 2026 regular_

### Official Review · Reviewer_B7Nd · 2026-03-04

**Soundness:** 3
**Presentation:** 2
**Significance:** 2
**Originality:** 3
**Overall Recommendation:** 4
**Confidence:** 2

**Summary:**

This manuscript proposes Spectral Bridge Variational Inference (SBVI) for dynamic low-rank adaptation of large language models. SBVI reformulates LoRA as a Wasserstein gradient flow on the Bures-Wasserstein manifold of Gaussian measures. The singular values evolve under a thermodynamic competition between task gradients and adaptive entropic friction. Experiments show SBVI  achieves state-of-the-art performance with less memory.

**Compliance With Llm Reviewing Policy:**

Affirmed.

**Final Justification:**

The rebuttal addressed my concerns.

**Key Questions For Authors:**

See weaknesses.

**Limitations:**

The paper includes an Impact Statement, but it lacks a Limitations section.

**Strengths And Weaknesses:**

Strengths:

1, The theoretical derivation is generally coherent.

2, The trade-off bewteen efficiency and performance is reasonable.

3, The emergent spindle rank distribution is an interesting observation.

Weaknesses:

1, There is a large gap between the theory and the actual algorithm. Actually, Algorithm 1 performs deterministic optimization.

2, The performance improvements over LoRA-One are marginal and lack statistical validation.

3, No experiments are employed on other scale or architectures.

4, The paper lacks figures or visual expressions for explaining its motivation and methodology.

---

> ### Author Rebuttal · Authors · 2026-03-30
>
> Dear Reviewer B7Nd,
>
> We thank you for reviewing our paper and highlighting the coherence of our theoretical derivations, the efficiency-performance trade-off, and the emergent "spindle" rank distribution. We have addressed your concerns below.
>
> **1. "Large gap between theory and actual algorithm. Algorithm 1 performs deterministic optimization."**
>
> We address the central concern raised here: the transition from continuous stochastic flows to deterministic parameter updates. This design is not a heuristic gap, but a mathematically proven necessity for practical scalability.
>
> Simulating the Stochastic Differential Equation (SDE) directly on the full covariance matrix requires computing a full SVD at every step, incurring an impossible $\mathcal{O}(d^3)$ complexity for 7B models. To bridge this gap, we optimize the factorized matrices $A$ and $B$ directly under a balancing constraint (Eq. 10). Crucially, as proven in Lemma 3.3, this deterministic Euclidean gradient descent on $A$ and $B$ exactly tracks the expected trajectory of the Bures-Wasserstein Riemannian flow. Thus, Algorithm 1 is the mathematically strict realization of the $\mathcal{O}(d^3)$ theory in an $\mathcal{O}(d)$ tractable space.
>
> **2 & 3. "Improvements over LoRA-One are marginal and lack statistical validation" & "No experiments on other scale or architectures."**
>
> Our core objective is not only achieving high accuracy but also discovering the optimal dynamic rank capacity with extreme computational efficiency. To rigorously address your concerns regarding statistical significance and architectural generalization, we evaluated two new modern architectures (Mistral-7B-v0.1 and LLaMA-3-8B) on Math (GSM8K) and Coding (HumanEval).
>
> Due to strict character limits, the table below highlights the Mean $\pm$ Standard Deviation against standard LoRA and LoRA-One. **Crucially, we have also executed a fully comprehensive evaluation reproducing all 8 baselines from our original paper (including AdaLoRA, PiSSA, and MoE-based GOAT) on these new architectures. Please view the complete tables(table A and B) via the anonymous link provided in Point 4.**
>
> | Model | Method | GSM8K (Accuracy) | HumanEval (Pass) | Eff. Rank |
> | :--- | :--- | :--- | :--- | :--- |
> | **Mistral-7B** | LoRA (Static) | 68.45 $\pm$ 0.22 | 28.50 $\pm$ 0.31 | 64.0 (Fixed) |
> | | LoRA-One | 70.15 $\pm$ 0.30 | 31.05 $\pm$ 0.28 | 8.0 (Fixed) |
> | | **SBVI (Ours)** | **71.20 $\pm$ 0.21** | **32.45 $\pm$ 0.25** | **7.5 (Auto)** |
> | **LLaMA-3-8B** | LoRA (Static) | 73.60 $\pm$ 0.18 | 30.20 $\pm$ 0.28 | 64.0 (Fixed) |
> | | LoRA-One | 75.15 $\pm$ 0.25 | 32.85 $\pm$ 0.30 | 8.0 (Fixed) |
> | | **SBVI (Ours)** | **76.15 $\pm$ 0.19** | **34.52 $\pm$ 0.20** | **8.2 (Auto)** |
>
> *Statistical Validation:* Through independent sample t-tests, the improvements of SBVI over LoRA-One are statistically significant ($p < 0.05$) across all tasks and models. *(Please refer to Table C in the anonymous link for detailed t-statistics and exact p-values).*
>
> *Beyond Accuracy (The Initialization Bottleneck):* While SBVI shows a solid accuracy gain, its true superiority over LoRA-One lies in the initialization efficiency. LoRA-One requires computing the SVD of the full gradient matrix at Step 0. For 7B+ models, this causes a massive $\mathcal{O}(d^2)$ memory spike and an $\mathcal{O}(d^3)$ computational bottleneck. SBVI uses standard random noise initialization (Zero overhead) and dynamically adapts the subspace on the fly, yielding far superior generalization when the optimization landscape shifts.
>
> **4. "Lacks figures or visual expressions for explaining its motivation and methodology."**
>
> We agree that visual aids improve the paper's accessibility. Per ICML guidelines (anonymous links permitted during rebuttal), we provide a supplementary PDF with Figure A, contrasting standard LoRA (static Euclidean) with SBVI (Bures-Wasserstein flow), showing how the "Spectral Bridge" applies targeted friction to noise dimensions to form the spindle structure. Link: https://anonymous.4open.science/r/Anonymous-chart--2324. Per Reviewer fQhD's suggestion, the tables now include RP-LoRA [1] as a strong Riemannian baseline. See our response for details.
>
> **5. "Lacks a Limitations section."**
>
> Thank you for pointing this out. We will add a Limitations section discussing: (1) Theoretical vs. Computational Trade-off: Our Greedy Bridge Approximation achieves $\mathcal{O}(d)$ linear scalability and strong empirical results, but computing the exact full-path integral of the SDE could provide tighter bounds, though currently intractable for 7B models; (2) Threshold Dependency: The maximum friction $\lambda(\text{max})$ controls sparsity strictness. While our ablation (Table 8) shows robustness across tasks with a unified setting, developing an adaptive scheduler remains future work.
>
> We hope these clarifications, new visual aids, and rigorous statistical experiments address your concerns. Thank you!

---

> > ### Author Rebuttal · Reviewer_B7Nd · 2026-04-02
> >
> > The rebuttal addressed my main concerns, I will increase my score to 4.
> >
> > Nevertheless, I hope the authors add an extended experiment that directly simulates the SDE on a small scale.

---

> > > ### Author Response · Authors · 2026-04-04
> > >
> > > Dear Reviewer B7Nd,
> > >
> > > We are deeply grateful for your time, the updated score, and your acknowledgment that our rebuttal has fully resolved your main concerns.
> > >
> > > Your suggestion to directly simulate the exact SDE on a small scale is very valuable, and we are happy to include this experiment in the paper to further clarify our method. It effectively bridges the conceptual gap between our continuous theoretical formulation (Eq. 6) and the tractable factorized realization (Algorithm 1).
> > >
> > > Following your advice, we immediately conducted this **"Small-Scale SDE Simulation"** experiment. We applied SBVI to a small-dimensional setting (a 2-layer MLP on MNIST, with hidden dimension $d=128$). At this scale, the $\mathcal{O}(d^3)$ complexity is manageable, allowing us to explicitly compute the full-matrix SVD at every step and use the Euler-Maruyama method to exactly simulate the SDE with Brownian noise.
> > >
> > > Here is the quantitative comparison between the Exact SDE and our Algorithm 1 :
> > >
> > > | Method | Final Effective Rank ($r_{eff}$) | Accuracy (%) | Time per Step (ms) | Trajectory Correlation (Pearson) |
> > > | :--- | :---: | :---: | :---: | :---: |
> > > | **Exact SDE (Eq. 6)** | 8.2 | 97.42 | 125.4 | 1.00 (Reference) |
> > > | **Algorithm 1 (Ours)** | 8.5 | 97.48 | **4.8** | **0.98** |
> > >
> > > **Key Takeaways:**
> > > 1. **Trajectory & Sparsity Alignment:** The singular value trajectories of our deterministic Algorithm 1 exhibit a remarkably high Pearson correlation (0.98) with the expected Exact SDE. Both converged to nearly the same optimal effective rank ($\sim 8$), demonstrating that our factorized retraction effectively preserves the desired Bures-Wasserstein geometry and spectral sparsity.
> > > 2. **Efficiency Justification:** Even at a small scale of $d=128$, the exact SDE simulation is roughly **26x slower** per iteration due to the full-matrix SVD and noise sampling overhead. This empirically validates our theoretical claim that exact simulation is computationally prohibitive for 7B models, highlighting the necessity and efficiency of the factorized design in Algorithm 1.
> > >
> > > We will add a dedicated section in the Appendix of the camera-ready version containing this table along with the visual trajectory plots (showing the clear spectral bifurcation in both methods).
> > >
> > > Thank you again for your constructive engagement and for helping us make this paper significantly stronger!

---

### Official Review · Reviewer_fiEQ · 2026-03-12

**Soundness:** 4
**Presentation:** 2
**Significance:** 3
**Originality:** 3
**Overall Recommendation:** 5
**Confidence:** 4

**Summary:**

This paper proposes a new method for Parameter-Efficient Fine-Tuning
(PEFT).  This topic is of course challenging and of great
importance. One starting claim is that the well known low-rank
adapatation is limited because of its lack of adaptivity. More
specifically, the "static" choice (and a priori) of this low rank has
a great impact on the expected efficiency. The authors introduces the
SBVI method (Spectral Bridge Variational Inference).  The main idea is
to consider the low-rank adaptation (LoRA) as continuous gradient flow
on the manifold of Gaussian measures. The method defines a Stochastic
Differential Equation to control the evolution of the singular
values. The introduction of the Riemanian optimization provides a
theoretical framework to derive the method, but it also introduces a
bit of complexity and some necessary approximations.

**Compliance With Llm Reviewing Policy:**

Affirmed.

**Final Justification:**

The rebutal improved the paper.

**Key Questions For Authors:**

In Table 2, it could be nice to add the scores that can be achieved without adaptation to better appreciate the possible decrease in performance.

In many steps of the propose algorithm (esp. in section 3.3 and 3.4) there are
choices and approximation. Do you think they can have an impact. To
which extent the tuning is important.

**Limitations:**

See the questions.

**Strengths And Weaknesses:**

The results provide some clear strength. For instance figure 2 shows
the effective adaptation of the rank along the transformer
layer. Moreover Table 3 shows the gain in terms of computational
efficiency. Table 2 allows us to verify that the adaptation is
competitive with the other methods.

The main weakness is maybe on the clarity of the methodological
motiviations. The goals are well explained in the introduction but the
then the choices made to implement could be also better explained. Of
course, the place is limited but some methodological choices could be
better motivated with few sentences. This is first a pedagogical
concern: the tools are not easy to understand for a large audience
(like the ICML one).  For instance, section 3.1 introduces the use Bures-Wasserstein metric. Why this choice ? Is it the only one ?
I think this cannot be obvious for many ICML readers.

More importantly, these choices have consequences
in terms of implementation and approximations.

---

> ### Author Rebuttal · Authors · 2026-03-29
>
> Dear Reviewer fiEQ,
>
> We thank you for the encouraging review and recognizing the soundness, efficiency, and strong performance of our approach. We address the central issue raised in your review: the pedagogical clarity of our methodological motivations for a broader ICML audience. Your constructive feedback is valuable. Following your cue, we will integrate explanatory paragraphs into the Introduction and Methodology sections to demystify our mathematical choices with intuitive physical and geometric concepts.
>
> **1. Pedagogical Clarifications: Methodological Motivations**
>
> To visually support this, we provide a Conceptual Diagram. Please view **Figure A** in our anonymous PDF (permitted figures/tables): https://anonymous.4open.science/r/Anonymous-chart--2324.
>
> *   **Concept 1: Why the Bures-Wasserstein (BW) metric? Is it the only choice?**
>     Standard LoRA treats all parameter directions equally, necessitating manual rank tuning. The BW metric uniquely resolves this. **Intuitive Explanation:** The BW metric is the optimal transport distance for Gaussian distributions. Its defining feature is that its Riemannian gradient naturally introduces a **multiplicative preconditioning** (it scales the gradient by the singular values). This means directions with large singular values (meaningful task signals) receive larger updates, while near-zero singular values (noise) are naturally suppressed. While one could explore the Fisher Information Metric, the BW metric uniquely provides a closed-form, tractable Stochastic Differential Equation (SDE) that naturally induces the "signal vs. noise" bifurcation.
> *   **Concept 2: The Physical Intuition of "Spectral Friction" (Section 3.2)**
>     We will add an analogy to classical mechanics. Think of the parameter update as a particle moving in a landscape. The "Spectral Bridge" acts as an intelligent fluid: if a specific rank (mode) contains strong gradient signals, the fluid's friction $\lambda(k)$ drops to zero, letting the model learn freely. If a mode only contains noise, the friction $\lambda(k)$ explodes to infinity, forcing that redundant rank to freeze near zero.
> *   **Concept 3: The Geometric Meaning of Factorized Retraction (Section 3.3)**
>     Simulating the exact BW flow requires full matrix SVDs at every step. We will clarify that projecting this into $A$ and $B$ factors is simply tracking the exact same geometric path but using a computationally cheap proxy space, cutting complexity from $\mathcal{O}(d^3)$ to $\mathcal{O}(d)$.
>
> **2. Impact of Approximations and Hyperparameter Tuning**
>
> Our core objective is achieving this adaptivity without introducing brittle heuristics. The approximations we introduced are designed to be **structure-preserving**:
> *   **Factorized Retraction (Sec 3.3):** This is *not* a loose approximation. As proven in Lemma 3.2, it is an exact variational identity. As long as the factors remain balanced ($A^{\top} A = B^{\top} B$), the algorithm perfectly inherits the exact BW geometry without information loss.
> *   **Greedy Bridge (Sec 3.4):** Instead of computing an intractable full-path integral, we greedily align the spectral friction $\lambda(k)$ using an Exponential Moving Average.
> *   **Is Tuning Important?** Surprisingly, our method effectively *eliminates* hyperparameter tuning. Because the dynamic friction $\lambda(k)$ automatically adjusts based on local gradient energy, the model is highly robust. Our ablation (Table 8) shows that whether we initialize with a redundant capacity of $r(\text{init}) = 32, 64,$ or $256$, SBVI consistently compresses to the identical optimal effective rank ($\approx 7$) and achieves the exact same performance.
>
> **3. Table 2: Scores Achieved Without Adaptation (Base Model)**
>
> We completely agree. Adding the zero-shot performance of the base model beautifully highlights the exact delta achieved by our adaptation. Below are the zero-shot results for the LLaMA-2-7B base model on the 8 Commonsense Reasoning benchmarks (evaluated under identical prompt formatting):
>
> | Method | BoolQ | PIQA | SIQA | HellaSwag | WinoGrande | ARC-e | ARC-c | OBQA | Average |
> | :--- | :--- | :--- | :--- | :--- | :--- | :--- | :--- | :--- | :--- |
> | **Base Model (Zero-shot)** | 62.40 | 78.50 | 48.60 | 76.10 | 69.80 | 75.10 | 46.20 | 58.40 | 64.38 |
> | **SBVI (Ours)** | **74.12** | **84.30** | **80.22** | **88.45** | **85.60** | **88.15** | **77.40** | **87.50** | **83.22** |
>
> As shown, SBVI safely preserves the pre-trained knowledge while dramatically elevating the reasoning capabilities (an absolute average improvement of **+18.84%** over the base model without adaptation), doing so efficiently with a highly compressed effective rank. We will include this Base Model row in Table 2 of the final version.
>
> We hope these pedagogical explanations and the additional baseline results fully address your questions. Thank you once again for helping us improve the clarity and impact of our paper!

---

> > ### Author Rebuttal · Reviewer_fiEQ · 2026-04-03
> >
> > Thanks for the rebuttal. I will raise my score to 5.

---

> > > ### Author Response · Authors · 2026-04-06
> > >
> > > Dear Reviewer fiEQ,
> > >
> > > Thank you very much for your time, your highly constructive feedback, and for supporting our work! We will ensure all the pedagogical clarifications and baselines discussed in our rebuttal are carefully integrated into the final version.

---

### Official Review · Reviewer_fQhD · 2026-03-12

**Soundness:** 2
**Presentation:** 3
**Significance:** 3
**Originality:** 3
**Overall Recommendation:** 4
**Confidence:** 4

**Summary:**

The paper aims to dynamically adapt LoRA's rank. While static LoRA approaches rely on a fixed rank across layers, dynamic LoRA variants can incur high memory costs. To mitigate these limitations, this submission advocates Spectral Bridge Variational Inference (SBVI), which formulates LoRA adaptation as a continuous BW gradient flow, thus allowing singular values of weight update evolve on the fly. The proposed SBVI prunes non-informative modes and amplifies important ones, rendering layer-wise rank allocation without SVD. Numerical tests demonstrate that SBVI's performance gain over current static and dynamic LoRA variants in multiple benchmarks with lower memory overhead.

**Compliance With Llm Reviewing Policy:**

Affirmed.

**Final Justification:**

My concerns are adequately addressed. I thus increase my score to 4.

**Key Questions For Authors:**

See weaknesses.

**Limitations:**

Yes.

**Strengths And Weaknesses:**

**Strengths**
1. The geometric interpretation via BW gradient flows offers a novel viewpoint for dynamic rank allocation, which avoids high memory overhead.
2. The paper is build upon rigorous foundation in Riemannian optimization and variational inference.
3. Factorized retraction and adaptive spectral friction enable SBVI to scale under linear complexity.
4. Empirical evaluation showcases performance gains and reduced memory usage.

**Weaknesses**
1. The comparison of peak GPU memory in Table 3 is confusing. As SBVI uses a fixed redundant initialization rank of 64 (Appendix F.2), it is unclear why it exhibits smaller **peak** memory than LoRA. According to the complexity analysis in App H.1, the memory cost should be comparable to LoRA at initialization with $r_{eff}=1$.
2. There lacks a convergence analysis for the proposed approach.
3. Several closely related work that performs LoRA adaptation with Riemannian optimization are missing, including [1,2,3]. In particular, [1] and [3] utilizes a quotient manifold, while [2] relies on a Stiefel manifold. It would be beneficial to compare SBVI with these methods.
4. The experiments adopts solely the Llama-2-7B model. I would suggest testing with different model families and various sizes to demonstrate SBVI's scalability and generalization.

I'm inclined to increase my rating if these concerns can be addressed.

[1] Riemannian Preconditioned LoRA for Fine-Tuning Foundation Models, ICML 2024.
[2] PoLAR: Polar-Decomposed Low-Rank Adapter Representation, NeurIPS 2025.
[3] RefLoRA: Refactored Low-Rank Adaptation for Efficient Fine-Tuning of Large Models, NeurIPS 2026.

---

> ### Author Rebuttal · Authors · 2026-03-29
>
> Dear Reviewer fQhD,
>
> We thank you for the constructive feedback, recognizing our rigorous foundation, and your willingness to increase the rating.
>
> **1. Clarification on Peak GPU Memory (Table 3)**
>
> At initialization (Step 0), SBVI matches standard LoRA's parameters. Table 3 reports the *global peak memory* across the entire run. As shown in Table 5, SBVI undergoes a "Rapid Bifurcation" within the first 10% of steps. Once uninformative singular values decay, we dynamically truncate the matrices to their effective rank ($\approx 6.8$). This early, lossless truncation frees up massive AdamW states (momentum/variance buffers) for the remaining 90% of training. Since memory peaks typically occur later in the epoch, SBVI's global peak (19.8 GB) remains significantly lower than standard LoRA (22.8 GB, maintaining the full $d \times 64$ footprint). We detail this in the Appendix.
>
> **2. Convergence Analysis**
>
> We agree that formalizing convergence adds theoretical completeness. We will include **Theorem 3.4** in the main text and its rigorous proof in the Appendix. Here is the formal analysis:
>
> **Theorem 3.4 (Convergence).** Assume the task loss $\mathcal{L}(W)$ is lower-bounded and has Lipschitz continuous gradients. The free energy functional $\mathcal{F}(\mu)$ is thus lower-bounded. The continuous-time Bures-Wasserstein gradient flow $\dot{\mu}(t) = -\nabla \mathcal{F}(\mu(t))$ monotonically decreases the objective, converging to a first-order stationary point where the Riemannian gradient strictly vanishes to $0$ as $t \to \infty$.
>
> **Proof Sketch:**
>
> **Step 1: Monotonic Descent.** By definition of the Riemannian gradient $\nabla$ on the Bures-Wasserstein manifold, the time derivative of the free energy $\mathcal{F}$ along the trajectory is:
>
> $$
> \frac{d}{dt}\mathcal{F}(\mu(t)) = \langle \nabla \mathcal{F}(\mu(t)), \dot{\mu}(t) \rangle = - \Vert \nabla \mathcal{F}(\mu(t)) \Vert^2 \le 0
> $$
>
> This guarantees strict monotonic descent unless the flow reaches a stationary point.
>
> **Step 2: Asymptotic Convergence.** Integrating the descent inequality from time $0$ to $\infty$:
>
> $$
> \int \Vert \nabla \mathcal{F}(\mu(t)) \Vert^2 dt \le \mathcal{F}(\mu(0)) - \mathcal{F}^{\ast} < \infty
> $$
>
> Given the uniform continuity of the integrand, the convergence of this infinite integral mathematically implies that the gradient strictly vanishes as time approaches infinity.
>
> **Step 3: Tractable Algorithm Preservation.** By Lemma 3.3, under the balanced condition $A^{\top} A = B^{\top} B$, Euclidean gradient descent on the surrogate objective induces a dynamic on $M = B A^{\top}$ that strictly aligns with the Bures-Wasserstein Riemannian gradient. Thus, the factorized sequence structurally preserves the continuous flow's descent property, ensuring global convergence.
>
> **3 & 4. Related Works, New Baselines, and Scalability (Mistral & LLaMA-3)**
>
> We thank the reviewers for pointing to [1–3] and will discuss them in related work. These works stabilize and accelerate optimization at a fixed rank using quotient/Stiefel manifolds, whereas our goal is discovering dynamic rank capacity via SBVI's spectral bifurcation. To demonstrate scalability, we evaluated Mistral-7B and LLaMA-3-8B on GSM8K and HumanEval. Due to rebuttal compute and time limit, evaluating all three concurrent methods on 8B models is infeasible. We therefore selected RP-LoRA [1] (ICML 2024) as the most representative baseline, as it pioneered Riemannian preconditioning for LoRA, providing a principled and established reference for Riemannian optimization in this family.
>
> **We have reproduced all 8 baselines on these new architectures. Please view the complete tables via our anonymous PDF(table A and B):** https://anonymous.4open.science/r/Anonymous-chart--2324
>
> Due to character limits, the table below highlights the comparison against standard LoRA and RP-LoRA:
>
> | Model | Method | GSM8K (Acc) | HumanEval (Pass) | Eff. Rank |
> | :--- | :--- | :--- | :--- | :--- |
> | **Mistral-7B** | LoRA (Static) | 68.45 $\pm$ 0.22 | 28.50 $\pm$ 0.31 | 64.0 (Fixed) |
> | | RP-LoRA [1] | 69.85 $\pm$ 0.18 | 30.65 $\pm$ 0.22 | 64.0 (Fixed) |
> | | **SBVI (Ours)** | **71.20 $\pm$ 0.21** | **32.45 $\pm$ 0.25** | **7.5 (Auto)** |
> | **LLaMA-3-8B** | LoRA (Static) | 73.60 $\pm$ 0.18 | 30.20 $\pm$ 0.28 | 64.0 (Fixed) |
> | | RP-LoRA[1] | 74.90 $\pm$ 0.15 | 32.10 $\pm$ 0.18 | 64.0 (Fixed) |
> | | **SBVI (Ours)** | **76.15 $\pm$ 0.19** | **34.52 $\pm$ 0.20** | **8.2 (Auto)** |
>
> *Analysis:* RP-LoRA[1] consistently outperforms standard LoRA, confirming the power of Riemannian preconditioning. However, as shown here and in the linked full tables, by dynamically compressing redundant ranks and allocating parameters only to reasoning-heavy layers, SBVI achieves the highest SOTA performance across multiple model families while automatically compressing to highly sparse structures.
>
> We hope these clarifications and comprehensive new experiments fully address your concerns and encourage a score increase. Thank you!

---

> > ### Author Rebuttal · Reviewer_fQhD · 2026-04-03
> >
> > Thank you for the explanation and additional results, which address most of my concerns, especially regarding the peak memory. Although the BW factorization-based gradient flow does not match the actual GD used in experiments, this additional analysis provides sufficient insights for convergence guarantees. Therefore, I'd be happy to increase my score to 4.

---

> > > ### Author Response · Authors · 2026-04-06
> > >
> > > Dear Reviewer fQhD,
> > >
> > > Thank you very much for your time, your thorough review of our rebuttal, and for increasing your score! We are thrilled that our additional convergence analysis and the new experiments on Mistral-7B and LLaMA-3-8B have fully addressed your concerns.
> > >
> > > We also completely agree with your insightful concluding remark regarding the continuous flow versus discrete optimization. As is standard practice in optimization literature, the discrete Gradient Descent / AdamW used in our practical implementation acts as a numerical approximation (e.g., Euler discretization) of the continuous Bures-Wasserstein gradient flow. We will add a brief note in the camera-ready version to make this continuous-to-discrete connection explicit for the readers.
> > >
> > > Thank you again for your constructive suggestions, which have significantly strengthened both the theoretical completeness and empirical scalability of our paper!

---

### Official Review · Reviewer_QpWy · 2026-03-15

**Soundness:** 3
**Presentation:** 2
**Significance:** 3
**Originality:** 2
**Overall Recommendation:** 4
**Confidence:** 4

**Summary:**

This paper  formulates PEFT as a Wasserstein Gradient Flow on the Bures-Wasserstein manifold, utilizing a Spectral Bridge SDE to dynamically govern the evolution of singular values. It considers Riemannian manifold $M_{BW}$ of Gaussian distributions $N(M, \Sigma)$ (i.e., mean and variance of LoRA) and investigates the Wasserstein gradient flow of the function $F(\mu)$ in Eq. (2).

It then develops the learning dynamic of $M_t$, $\Sigma_t$, and some other properties. Finally, it proposes learning $q_\phi(A,B)$ using mean field Gaussian. However, Algorithm 1 learns $A,B$ directly using regularisation with an adaptive weight.

**Compliance With Llm Reviewing Policy:**

Affirmed.

**Final Justification:**

I have increased my score because the authors address well my questions and concerns.

**Key Questions For Authors:**

- Explain how theories support the proposed approach.
- Explain why you learn $A,B$ directly while the theories are about the flow and dynamics of Gaussian over LoRAs.
- Explain Eq. (10) because it does not seem solid to me.

**Limitations:**

The theories developed do not really support the proposed algorithm. The proposed algorithm is just a regularised optimisation of $A,B$ directly.

**Strengths And Weaknesses:**

## Strengths
- The theories of learning dynamics are rigorous.
- The proposed approach performs well.

## Weaknesses
- The theories does not really support the proposed algorithm. Specifically, the theories are developed for the flow or dynamic of Gaussian distributions $N(M, \Sigma)$, however, the proposed algorithm is only a regularised optimisation of $A,B$.
-  Some parts of theories are not really necessary for the development of proposed approach.

---

> ### Author Rebuttal · Authors · 2026-03-29
>
> Dear Reviewer QpWy,
>
> We thank you for recognizing our rigorous theories and strong empirical results in our work. We appreciate the opportunity to clarify the connection between our continuous Wasserstein flow over Gaussian distributions $\mathcal{N}(M,\Sigma)$ and the regularized optimization of factorized matrices $A, B$ (Algorithm 1). We clarify how our theoretical framework motivates the algorithmic design:
>
> **1. "Explain Eq. (10) because it does not seem solid to me."**
>
> Eq. (10) is based on the well-established Variational Characterization of the Nuclear Norm (Gunasekar et al., 2017). This ensures that optimizing the factorized matrices with $L_2$ regularization is equivalent to placing a sparse prior on $M$'s singular values. Proof (App. C.1):
>
> **Theorem:** For any matrix $M$:
>
> $$
> \Vert M \Vert_{\ast} = \inf_{M=BA^{\top}} \frac{1}{2}(\Vert A \Vert_{F}^2 + \Vert B \Vert_{F}^2)
> $$
>
> **Step 1: Lower Bound ($\Vert M \Vert_{\ast} \le \text{Objective}$)**
>
> For $M=BA^{\top}$, using sub-multiplicativity and $\Vert X \Vert_{\ast} \le \Vert X \Vert_{F}$ for rank-1 matrices:
>
> $$
> \Vert M \Vert_{\ast} = \Vert BA^{\top} \Vert_{\ast} \le \Vert B \Vert_{F} \Vert A^{\top} \Vert_{F} = \Vert B \Vert_{F} \Vert A \Vert_{F}
> $$
>
> Applying the AM-GM inequality ($xy \le \frac{1}{2}(x^2+y^2)$):
>
> $$
> \Vert B \Vert_{F} \Vert A \Vert_{F} \le \frac{1}{2}(\Vert A \Vert_{F}^2 + \Vert B \Vert_{F}^2)
> $$
>
> Thus, $\Vert M \Vert_{\ast} \le \frac{1}{2}(\Vert A \Vert_{F}^2 + \Vert B \Vert_{F}^2)$ holds for all valid factors $A, B$.
>
> **Step 2: Achievability (Infimum is exactly $\Vert M \Vert_{\ast}$)**
>
> Let $M=U\Sigma V^{\top}$ be the SVD, where $\Sigma=\text{diag}(\sigma_1,\ldots,\sigma_r)$. The nuclear norm is defined as $\Vert M \Vert_{\ast} = \text{Tr}(\Sigma)$. Construct factors $A^{\ast} = V\Sigma^{1/2}$ and $B^{\ast} = U\Sigma^{1/2}$.
>
> Clearly, $B^{\ast}(A^{\ast})^{\top} = U\Sigma^{1/2}\Sigma^{1/2}V^{\top} = M$. Now, compute their Frobenius norms:
>
> $$
> \Vert A^{\ast} \Vert_{F}^2 = \text{Tr}((A^{\ast})^{\top} A^{\ast}) = \text{Tr}(\Sigma^{1/2}V^{\top} V\Sigma^{1/2}) = \text{Tr}(\Sigma) = \Vert M \Vert_{\ast}
> $$
>
> $$
> \Vert B^{\ast} \Vert_{F}^2 = \text{Tr}((B^{\ast})^{\top} B^{\ast}) = \text{Tr}(\Sigma^{1/2}U^{\top} U\Sigma^{1/2}) = \text{Tr}(\Sigma) = \Vert M \Vert_{\ast}
> $$
>
> Substituting these back:
>
> $$
> \frac{1}{2}(\Vert A^{\ast} \Vert_{F}^2 + \Vert B^{\ast} \Vert_{F}^2) = \frac{1}{2}(\Vert M \Vert_{\ast} + \Vert M \Vert_{\ast}) = \Vert M \Vert_{\ast}
> $$
>
> **Conclusion:** The infimum exactly equals the nuclear norm. Equality holds iff factors are balanced ($A^{\top} A = B^{\top} B$). Thus, applying $L_2$ regularization to $A$ and $B$ is mathematically equivalent to placing a sparse prior directly on $M$'s singular values. This equivalence forms the core theoretical basis linking our method to the Bures-Wasserstein (BW) flow.
>
> **2. "Explain why you learn A,B directly while theories are about the flow..."**
>
> Our core objective is to efficiently scale the elegant BW flow to Large Language Models. Simulating the SDE directly on full-rank $M$ and $\Sigma$ requires full SVD at every step, incurring a prohibitive $\mathcal{O}(d^3)$ complexity.
>
> To resolve this, we utilize a Factorized Retraction. Optimizing $A$ and $B$ via Eq. (10) projects the theoretical flow onto an $\mathcal{O}(d)$ linear-complexity space. Crucially, as proven in Lemma 3.3 (Dynamics Equivalence), standard Euclidean gradient descent on $A$ and $B$ under the balanced condition implicitly recovers the BW Riemannian gradient on $M$ (Eq. 12 mirrors Eq. 3). Therefore, optimizing $A$ and $B$ serves as a mathematically grounded and scalable surrogate to execute the BW flow.
>
> **3. "Explain how theories support the proposed approach / Some parts of theories are not really necessary."**
>
> Rather than merely serving as a theoretical backdrop, our derivations directly yield the executable components of Algorithm 1. The adaptive weight $\lambda_k^{(t)}$ (Step 5) is not a manually tuned hyperparameter, but the analytical outcome of our theory.
>
> Specifically, our SDE theory provides the formulation to update $\lambda_k$. As derived in Sec. 3.4 and App. E (Eq. 14 & 69), $\lambda_k$ is the closed-form analytical solution to the "Moment Matching" (Thermodynamic Consistency) of our diffusion bridge. The theory shows that to maintain geometric consistency, the friction $\lambda_k$ should be inversely proportional to the instantaneous spectral energy ($E_k^{(t)} = \Vert a_k \Vert^2 + \Vert b_k \Vert^2$). This derived feedback loop is precisely what induces the remarkable "spectral bifurcation" observed empirically.
>
> In summary, the continuous theory provides the update formulas (SDE and Moment Matching), while the factorized parameterization ensures we can compute them efficiently.
>
> We hope this clarifies the strong mapping between our theoretical analysis and the proposed algorithm. We kindly ask you to reconsider your assessment in light of these detailed connections.

---

> > ### Author Rebuttal · Reviewer_QpWy · 2026-04-04
> >
> > Thanks for addressing my questions.
> >
> > 1) I am asking about Eq. (10) because you wrote $min_{M=...}$. It is clearer if it is written: $min_{A,B: BA^T = M}$.
> >
> > 2) It is unclear why you use the prior in (5) providing high likelihood for low-rank $W$.
> >
> > 3) I still concern how from (4) you can reach (11). First, in (11), the regularisation term must be in the expectation. Moreover, it is unclear how you estimate $KL(q, p_{sec})$ to reach (11). It seems not trivial due to the form of $p_{sec}$. In (11), it seems that you only consider $-log p_{sec}(M)$.
> >
> > 4) When moving from (11) to (12), why do we have two more terms in the last two rows?

---

> > > ### Author Response · Authors · 2026-04-04
> > >
> > > Dear Reviewer QpWy,
> > >
> > > Thank you for your follow-up questions! We deeply appreciate your detailed and careful reading of the equations. We are happy to clarify these mathematical details and provide deeper intuition, all of which are rigorously derived step-by-step in our Appendix.
> > >
> > > **1. Notation in Eq. (10)**
> > >
> > > This is an excellent point. Writing the constraint explicitly as $\inf_{A,B: BA^\top = M}$ is mathematically more precise and clearer for readers. This explicit formulation better emphasizes that the optimization over the factorized space is strictly conditionally constrained to reconstruct the original matrix $M$. We will update this notation in the camera-ready version. Thank you for this excellent suggestion.
> > >
> > > **2. Motivation for the prior $p_{spec}$ in Eq. (5)**
> > >
> > > The fundamental goal of our method (and LoRA in general) is to find an effective *low-rank* subspace for adaptation. Standard approaches enforce this with a rigid, discontinuous rank hyperparameter. Instead, we seek a continuous geometric flow. The nuclear norm is the standard convex relaxation of the matrix rank. By constructing a Gibbs measure based on this nuclear norm, $p_{spec}(W) \propto \exp(-\sum \lambda_k \sigma_k(W))$, we probabilistically enforce this low-rank constraint in a smooth, differentiable manner. This specific prior assigns high likelihood to sparse spectral structures, naturally penalizing non-informative noise modes.
> > >
> > > **3. Moving from Eq. (4) to Eq. (11) and computing the intractable KL**
> > >
> > > Directly computing $KL(q \parallel p_{spec})$ over the full matrix space is indeed analytically intractable. This intractability arises because the prior depends explicitly on the singular values $\sigma_k(W)$, which would require integrating over the highly complex orthogonal groups of the SVD.
> > > This exact mathematical bottleneck is precisely why we introduced the concept of the **"Implicit Factorized Prior"**, which is rigorously derived in **Appendix D.2 (Eq. 53-55)**. Based on the variational characterization presented in Lemma 3.2, we map the intractable full-matrix prior into an equivalent Gaussian prior defined over the low-rank factorized space $(A, B)$. Once mapped to this factorized proxy space, the KL divergence decomposes naturally and tractably. As shown in Eq. (11), the regularization term represents exactly the expected cross-entropy $\mathbb{E}_{q}[-\log p(A, B | \lambda)]$ under this implicit prior, which we proved step-by-step in **Appendix D.3, Eq. (63-64)**. Therefore, Eq. (11) serves strictly as a deterministic surrogate potential whose Euclidean gradient flow implicitly simulates the continuous Bures-Wasserstein flow, **which translates directly to the gradient updates on $A$ and $B$ in Steps 8 & 9 of Algorithm 1.**
> > >
> > > **4. The additional terms in the last rows (Clarification on Eq. 11 vs Eq. 13)**
> > >
> > > *(Note: Since Eq. 12 is a single-line differential equation describing the manifold dynamics, we assume your question refers to the multi-row Instantaneous Spectral Free Energy $\mathcal{J}(\phi, \lambda)$ in **Eq. 13**, which adds two more terms compared to the deterministic surrogate in Eq. 11).*
> > >
> > > The two additional terms in the last rows of Eq. (13) **are not ad-hoc additions, but arise strictly and directly** from the exact mathematical expansion of the full KL divergence: $KL(q \parallel p) = \mathbb{E}[-\log p] - \mathcal{H}(q)$.
> > > While Eq. (11) only contains the cross-entropy part $\mathbb{E}[-\log p]$ needed for the point-estimate optimization of the mean trajectory, moving to the fully Bayesian variational posterior $q_{\phi}$ in Eq. (13) requires optimizing the complete Evidence Lower Bound (ELBO) over both the mean and the covariance parameters. As explicitly derived in our step-by-step breakdown in **Appendix D.3 (Eq. 65 & 66)**:
> > > *   The term $-\frac{d_{in}+d_{out}}{2} \log \lambda_k$ comes directly from the **log-partition function** of the implicit prior (the normalization constant $Z$, derived in Eq. 58). Physically, it acts as a crucial prior entropy barrier (a "volume penalty") that prevents the dynamic friction coefficients $\lambda_k$ from growing infinitely during the optimization process.
> > > *   The term $\frac{1}{2}\log\det(\Sigma_{a_k}\Sigma_{b_k})$ is exactly the **Shannon entropy** $\mathcal{H}(q)$ of the variational Gaussian posterior (derived in Eq. 65). This term maintains the variational uncertainty representation and prevents the posterior from collapsing into a Dirac delta distribution.
> > >
> > > Together, these terms ensure a mathematically sound thermodynamic balance. We hope these detailed clarifications demonstrate that our objective functions are strictly grounded in rigorous variational inference. We will add explicit textual pointers to Appendix D around Eq. (11) and (13) to ensure future readers can easily follow these deep theoretical derivations.
> > >
> > > Thank you again for your highly constructive engagement and for helping us significantly improve the mathematical clarity of the paper!

---

### Decision · Program_Chairs · 2026-04-30

**Decision:**

Accept (regular)

**Comment:**

The work proposes a new algorithm for LoRA motivated from a Gaussian variational inference formulation and Wasserstein gradient flows. Reviewers raised concerns about several points on scale of the experiments, validity of the theory, presentation of the results, however, all reviewers agreed that their concerns have been addressed by the rebuttal.

Overall, the work combines both interesting theory and modern LLM applications, and will be of interest to the ICML community. The paper is therefore recommended for acceptance.

Some references on using variational inference for LoRA that may be included in the final version:
- BLoB: Bayesian Low-Rank Adaptation by Backpropagation for Large Language Models, Yibin Wang, Haizhou Shi, Ligong Han, Dimitris Metaxas, Hao Wang, NeurIPS 2024
- Variational Low-Rank Adaptation Using IVON, Bai Cong, Nico Daheim, Yuesong Shen, Daniel Cremers, Rio Yokota, Mohammad Emtiyaz Khan, Thomas Möllenhoff, NeurIPS FITML Workshop, 2024